# Stressors, manifestations and course of COVID-19 related distress among public sector nurses and midwives during the COVID-19 pandemic first year in Tasmania, Australia

Kathryn M. Marsden[1]*, I. K. Robertson[2], J. Porter[1]

**1** Tasmanian Health Service South, Hobart, Tasmania, Australia, **2** University of Tasmania, Tasmania, Australia

* kathryn.marsden@ths.tas.gov.au

## Abstract

Impacts of the COVID-19 pandemic on the mental health of healthcare workers has been established, linking workplace factors with high levels of stress, anxiety, depression, insomnia and burnout. Less established is how COVID-19 affects both work, home and social life of nurses and midwives concurrently. This study describes the prevalence and severity of anxiety, depression, post-traumatic stress disorder (PTSD) and insomnia and examines their associations with stressors within the work, home and social environment, among nurses and midwives. A longitudinal, mixed-methods, online survey explored the psychological health of public sector nurses and midwives during the COVID-19 pandemic first year. Surveys were conducted in April (initial) and June 2020 (3-month), and April 2021 (12-month) and consisted of psychological tests including the Patient Health Questionnaire, General Anxiety Disorder, Insomnia Severity Index, and the Impact of Events Scale-Revised; workplace and lifestyle questions, together with free-text comments. The relative strengths of the associations between predictor and outcome variables were estimated using repeated measures ordered logistic regression, and free text responses were themed. Data show diagnostic levels of anxiety (23%, 18%, 21%) at surveys one, two and three respectively, depression (26%, 23% and 28%), PTSD (16%, 12% and 10%) and insomnia (19%, 19% and 21%). The strongest predictors of psychological distress were current home and family stress and poor clinical team support. Factors which will help preserve the mental health of nurses and midwives include strong workplace culture, reducing occupational risk, clear communication processes, and supporting stable and functional relationships at home. The COVID-19 pandemic has increased the visibility of mental distress on nurses and midwives and established they are pivotal to healthcare. The health service has a duty-of-care for the welfare of nurses and midwives who have entered this psychologically taxing profession to future proof service delivery and safeguard its service-response capacity.

**Data Availability Statement:** All relevant data are within the paper and its Supporting Information

files. This includes the minimum data set and Additional Materials.

**Funding:** This research received funding from Florence Nightingale Grants, Tasmania, Australia. Clifford Craig Foundation provided in kind support for statistical support.

**Competing interests:** The authors have declared that no competing interests exist.

## Introduction

The deleterious effects relating to the COVID-19 pandemic on the mental health of nurses has now been widely established. An abundance of literature consistently reports the increased prevalence and severity of depression, anxiety, stress and sleep disturbance among nurses in Australia [1–4] and globally [5, 6]. International research relating to COVID-19 show one-third of healthcare workers have experienced greater psychological distress evidenced by nervousness, agitation, psychological fatigue and depression [5, 6] compared with the general population [7] and are at high risk for developing stress response syndromes, post-traumatic stress disorder (PTSD), chronic illness, and burnout [8]. Emotional exhaustion is a recognised symptom of burnout [9] which affects a person's ability to accomplish and sustain their professional role.

The COVID-19 pandemic posed many challenges and changes to daily life. Lockdowns and social restrictions to 'flatten the curve' or reduce direct COVID-19 exposure and transmission were also endured by nurses and midwives. Lifestyle changes such as lack of social, leisure and recreation activities impact on physical, mental and spiritual health and can precipitate psychological distress [10–12]. The dynamic effect of life satisfaction, financial uncertainty, social dysconnectivity, loneliness, isolation and the associated varied stages or levels of fear and vulnerability can overlie or augment precipitating stress factors [13].

The boundaries between work life and home and social life quickly became more permeable [14] for nurses and midwives with COVID-19 impacting immediate and distant home and family, work and employment networks. This created conflicting loyalties around commitment to safety for self, risk of viral transmission to family members and commitment to the profession and their employer [10]. COVID-19 challenges contagion fears, fears of adverse local and global economic consequences, xenophobia [15] and personal belief systems. Understanding how COVID-19 affects a nurse's work and home and social life is complex and multidimensional.

Previous studies on Severe Acute Respiratory Syndrome (SARS) and Ebola uncover the serious nature of psychological distress endured by healthcare workers, revealing the long-standing nature of the distress with many nurses suffering PTSD, depression, anxiety, and burnout persisting even after the outbreaks [16, 17]. We are learning that psychological distress from COVID-19, natural disasters [18] and other cumulative life stressors [19], also have long-standing effects [20] and it is often a few months down the track before psychological issues come to the fore [18]. Acute psychological distress can progress to a chronic state and for healthcare workers this can precipitate burnout [21].

Levels of psychological distress in the Australian general population show an estimated 13% with anxiety-related conditions, and an estimated 10% of the population suffering depression [22]. A small study prior to COVID-19 show 42% of Australian nurses suffering anxiety [23] and 41% with PTSD [24]. Also prior to the COVID-19 pandemic, it is reported 20% of Australian midwives suffering anxiety and 17% depression [24]. Comparatively during the COVID-19 pandemic, a large systematic review and meta-analysis in 2020 revealed a rate of anxiety of 37%, 35% depression and 43% PTSD among nurses [5]. Finally, data relating to insomnia show rates of 43% during the COVID-19 [5].

Nurses have always worked under intense psychological pressure day-to-day [10], long before the COVID-19 pandemic, yet little emphasis has ever been placed on understanding their wellbeing or psychosocial coping mechanisms. Nurses are pivotal to emergency pandemic preparedness and overall response capacity [10]. The additional, acute demands now being placed on nurses due to the current pandemic will arguably add to nurses' existing distress [25] compounding their risk of stress responses, post-traumatic stress disorder, anxiety,

depression, chronic illness, and burnout [26]. This longitudinal study describes the stressors and how they manifest in the context of the work, home and social environment.

## Study aims

1. Describe the level of anxiety, depression, insomnia and PTSD of nurses and midwives during the first year of the COVID-19 pandemic.

2. Determine the associations between the psychological outcomes and sociodemographic, workplace and home and social life predictors.

3. Examine common themes and characterise stressors and their manifestations of nurses and midwives during the first year of the COVID-19 pandemic.

## Methods

### Study design

This was designed as an on-line, longitudinal, mixed-methods, survey intending to explore the trajectory of the psychological health of nurses and midwives during the COIVD-19 pandemic first year in Tasmania, Australia. Due to the incomplete recording of unique personal identifiers by respondents, what happened was a hybrid design.

While the Tasmanian Health Service is a state-wide service, it is divided into three separate regions, South, North, and Northwest, each with separate, local research governance bodies. While overarching Ethics approval was gained from Human Research Ethics Committee (Tasmanian Network) Reference No. H0021677 on the 29th April 2020, researchers also required authority from regional local research governance committees prior to survey distribution.

Once approvals were gained, a survey was designed using SurveyMonkey® and a link emailed to public sector nurses and midwives on the Tasmanian Health Service email list at three timepoints between April 2020 and April 2021; Survey 1, (April 2020, initial), Survey 2, (July 2020, 3-month) and Survey 3 (April 2021, 12-month). Each survey was closed out after three weeks.

The Survey used is included in S1 File. Survey distribution was staggered, Survey 1 was circulated to public sector nurses and midwives in the Southern Tasmanian region only due to delays in local research governance processes in neighbouring regions. Survey 2 distribution was limited to the Southern and Northern regions only as communications were limited to essential communique across the Northwest region of Tasmania during April–July 2020 due to the critical nature of the COVID-19 outbreak at that time. Survey 3 was circulated across all Tasmanian regions.

Immediately prior to and during survey distribution, the study was advertised through means of email and posters. Information sheets were attached to the online survey. Completion of the online surveys implied consent. The survey was anonymous with respondents entering their own personal identifier on each survey intended for purposes of longitudinal analysis. Survey completion took approximately 20 minutes.

### Respondents

Estimate numbers based on health workforce planning data (2018) show 4884 nurses and midwives employed in the public sector in Tasmania. Breakdown of staff numbers per region show 2371 nurses and midwives employed in Southern Tasmania, 1638 Northern Tasmania, 874 North West Tasmania and 402 State-wide [27]. The numbers of respondents are shown in

**Table 1. Demographic characteristics of responders: Predictors, case numbers and missing data.**

| | Predictor | | Respondent numbers | | | | Missing |
|---|---|---|---|---|---|---|---|
| | | | Survey 1 | Survey 2 | Survey 3 | Total | |
| Q2 | Staff grade | All | 676 | 539 | 431 | 1646 | 30 |
| | State registered nurse | 1 | 572 | 473 | 362 | 1407 | |
| | State registered midwife | 2 | 40 | 26 | 26 | 92 | |
| | State enrolled nurse | 3 | 56 | 32 | 38 | 126 | |
| | Assistant in nursing | 4 | 8 | 8 | 5 | 21 | |
| | ["Others" allocated to the nearest similar employment class] | | | | | [44] | |
| Q3 | Covid ward status | All | 667 | 538 | 388 | 1631 | 45 |
| | Covid-negative patients | 0 | 547 | 487 | 388 | 1422 | |
| | Both | 1 | 41 | 20 | 13 | 74 | |
| | Covid-positive patients | 2 | 79 | 31 | 25 | 135 | |
| Q5 | Sector | All | 377 | 538 | 433 | 1348 | 328 |
| | Public | 1 | 359 | 517 | 412 | 1288 | |
| | Private/Both/Other | 0 | 18 | 21 | 21 | 60 | |
| Q6 | Gender | All | 670 | 538 | 433 | 1641 | 35 |
| | Female | 0 | 590 | 462 | 390 | 1442 | |
| | Male | 1 | 80 | 76 | 43 | 199 | |
| Q7 | Highest education level | All | 670 | 539 | 433 | 1642 | 34 |
| | Hospital certificate | 0 | 61 | 54 | 47 | 162 | |
| | TAFE certificate | 0 | 35 | 25 | 33 | 93 | |
| | Undergraduate student | 0 | 9 | 6 | 4 | 19 | |
| | Batchelor degree | 1 | 210 | 171 | 130 | 511 | |
| | Postgrad certificate/Masters/PhD | 2 | 355 | 283 | 219 | 857 | |
| Q8 | Age group (years) | All | 675 | 538 | 432 | 1645 | 31 |
| | 18–25 | 1 | 37 | 26 | 22 | 85 | |
| | 26–30 | 2 | 56 | 36 | 30 | 122 | |
| | 31–40 | 3 | 138 | 104 | 88 | 330 | |
| | 41–50 | 4 | 134 | 123 | 88 | 345 | |
| | 51–60 | 5 | 234 | 186 | 151 | 571 | |
| | 61–71 | 6 | 75 | 61 | 51 | 187 | |
| | 71+ | 7 | 1 | 2 | 2 | 5 | |
| Q9 | Social situation | All | 449 | 527 | 429 | 1405 | 271 |
| | Living with partner/family/friends | 1 | 373 | 434 | 352 | 1159 | |
| | Living caring for others | 2 | 65 | 78 | 69 | 212 | |
| | Living alone | 3 | 11 | 15 | 8 | 34 | |
| Q10 | Smoking | All | 664 | 491 | 392 | 1619 | 57 |
| | Non-smoker | 0 | 609 | 491 | 392 | 1492 | |
| | Smoker | 1 | 55 | 40 | 32 | 127 | |

the results section, Table 1. The timing of survey responses in relation to the occurrence of COVID-19 infections and hospitalisations in Tasmania are shown in Fig 1.

## Test battery

Currently there is no validated and standardised test battery to specifically evaluate the psychological outcomes of COVID-19 among nurses and midwives. A battery of tests was designed for online, survey assessment of a heterogeneous group of nurses and midwives. Study design was informed by the outcome literature and aimed to maintain a standardised approach as

much as possible. Psychological and psychosocial outcome domains were evaluated. Assessments consisted of standardised, subjective psychological screening tools, (validated for use in healthcare worker population where possible), sociodemographic, workplace and home and social life questions and open-ended questions which were themed to accurately reflect the participants' experiences.

### Psychological outcome measures

**Patient Health Questionnaire (PHQ-9).**   The PHQ-9 is a nine item self-screening tool for symptoms of depression based on the diagnose criteria of DSM IV [28]. The PHQ-9 has a dual-purpose to screen for the presence of a depressive disorder as well as to grade depressive symptom severity. The PHQ-9 score ranges from 0 to 27. It is based on the nine items scored from 0 "not at all" to 3 "nearly every day". Depression severity: 0–4 none, 5–9 mild, 10–14 moderate, 15–19 moderately severe, 20–27 severe. Cut-off sensitivity used in this study for detection of depression is $\geq$10.

**General Anxiety Disorder (GAD-7).**   The GAD-7 is a seven item self-screening tool for symptoms of general anxiety disorder and has been used across various settings and populations [29, 30] and has a range from 0–21. Anxiety severity ranges: 0–4 none to minimal, 5–7 mild (recommended to monitor symptoms), 8–9 mild though likely to be diagnosed with an anxiety disorder, 10–14 moderate symptoms are clinically significant, 15–21 severe symptoms warrant active treatment. A cut-off score of 10 has been identified as the optimal point for sensitivity (89%) and specificity (82%). Cut-off sensitivity used in this study for detection of anxiety is $\geq$10.

**Insomnia Severity Index (ISI).**   The ISI is a self-screening tool for symptoms of insomnia, consisting of seven items and assesses the nature, severity, and impact of insomnia. A five-point rating scale is used to rate each item, with 0 no problem and 4 very severe problem, yielding a total score ranging from 0 to 28. The total score is interpreted as: absence of insomnia (0–7), sub-threshold insomnia (8–14), moderate insomnia (15–21), and severe insomnia (22–28). A higher sensitivity cut-off of $\geq$10 was used in this study for detection of insomnia [31, 32].

**Impact of Events Scale-Revised (IES-R).**   The IES-R is a self-screening tool for symptoms of PTSD, providing a dimensional assessment of general stress and PTSD [33]. Participants specify the frequency with which they have had intrusion-, avoidance-, and hyperarousal-related thoughts in the previous seven days on a Likert scale. Total score interpretation as follows: PTSD is of clinical concern (24–32), probable diagnosis of PTSD (33–36) and scores 37 and above are high enough to supress your immune system's functioning. Current study analyses was performed using the total score only and a sensitivity cut-off for detection of PTSD of $\geq$33.

### Predictor variables

Details of the questionnaire responses, numbers of each response is shown in Section 2.2 page 6–11, and Table A3 in S3 File.

- Sociodemographic predictors include age, gender, education level, social situation, smoking status, job status

- Workplace predictors include COVID-19 exposure, access to PPE, access to COVID-19 testing, information and communications, training and preparation for redeployment, workplace team support, provision of quality patient care and tangible workplace support (food, accommodation, and transportation).

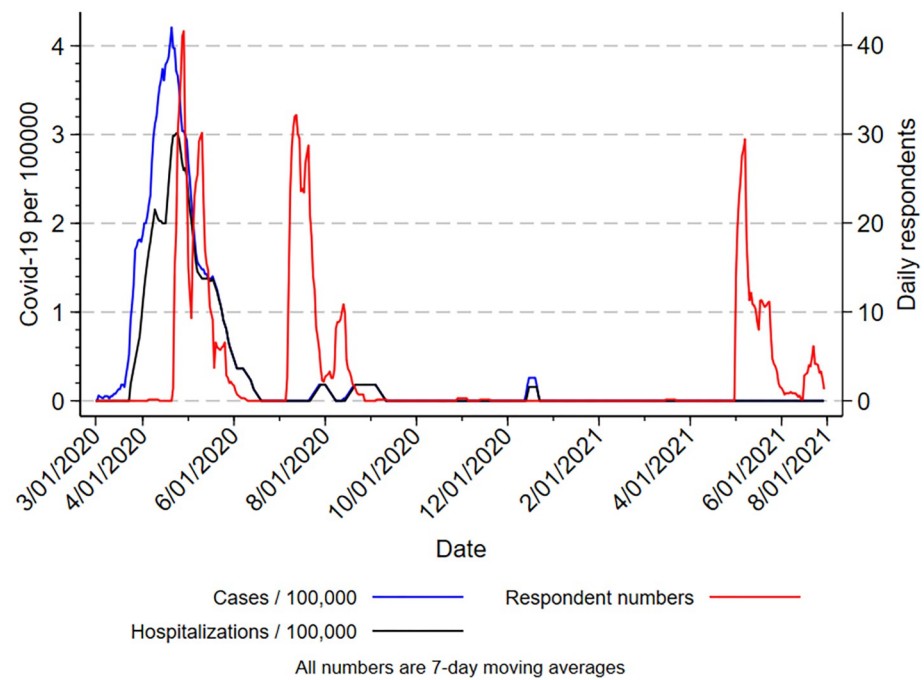

**Fig 1. Relationship between dates of occurrence of COVID-19 cases and hospitalisations in Tasmania and numbers of responses during the three survey periods.**

- Home and social life predictors include current home and family stress, future home and family stress, current financial stress, and future financial stress

Likert scales were used to classify these predictors, and the scales were rank ordered in nature. The scales differed, depending on the precise questions (see Survey in S1 File). Respondents were given options not to respond to the question, and these were treated as missing data.

Questions included, "To what degree has COVID-19 contributed to your current home and family stress? To what degree do you consider COVID-19 will contribute to your future home and family stress? To what degree has COVID-19 contributed to your current financial stress? To what degree do you consider COVID-19 will contribute to your future financial stress?" Participants were given the option to add free text comments. These comments were themed.

## Qualitative data analysis

Open ended survey questions were used to elicit participant's stories and perspectives relating to the impact of COVID-19 on predictors of psychological distress. This participant-centred approach complements the inventory of standardised criteria and symptom severity by deriving understanding directly from the participants' words. Collecting open-ended survey responses aimed to capture and understand the lived experience of COVID-19 and the participant variation of the overall experience. Narrative and language were informative in the absence of interviews and dialogue. This provided a snapshot of the language and descriptors used by the participant groups allowing some insight into emotions throughout the pandemic.

## Thematic analysis

Thematic analysis was used to ascertain themes from the participant open-ended survey questions responses, looking for patterns, meaning and potential points of interest in the data that

maintains an honest interpretation by constant reference to the raw data source [34]. Using both semantic and latent levels of thematic analysis, researchers interpreted the participants' words by looking beyond them to identify their meaning and implications.

Two researchers, each independently scrutinised the same data responses. Each response was coded separately and catalogued by survey period and question. Responses were systematically coded, extracting the most significant statements to explore for well-established and emergent themes. Second stage coding identified patterns from the initial first stage, until no new codes were found. During analysis meetings, independently formulated meanings were discussed until agreement was reached. The resultant development of principle-overarching themes and subthemes [35] was generated through a transparent process that can be traced back to the participants' words across multiple participant groups.

From this framework, response trajectories were formed. To create longitudinal trajectories of the response data, each response theme was placed into a trajectory framework [34–36] and categorised as 'positive', 'negative', 'mixed' or 'neutral.' Although we structured coding by using five themes and subthemes, scientific and clinical judgement and discussion among coders was a key part of the analytic process.

The condensed themes and subthemes include miscommunication with the subtheme of inadequate leadership; occupational risk with the subtheme vulnerability; disconnect with the subthemes of lack of social supports, economic uncertainty, blurred roles and permeable boundaries; workforce dissolution with the subthemes devalued and disengaged; positive life change with the subtheme certainty.

## Quantitative data analysis

The four primary psychological outcome scales (PHQ-9, GAD-7, ISI, IESR) are rank ordered with relatively broad ranges. The prevalence of the scores (percentage of respondents; exact binomial 95% confidence intervals (95%CI)) above thresholds for moderate and severe psychological diagnoses were estimated. The change per year of the prevalence estimates were estimated by Poisson regression: binomial outcomes (disorder present or absent) were regressed against the time from the start of Survey 1, and expressed as an incidence rate ratio (IRR; 95%CI; P-values).

Summary outcome scores were described as medians and inter-quartile ranges (IQR) in different subgroups. The relative strengths of the associations between outcome scores and predictor variables were assessed simultaneously with multivariate repeated-measures ordered logistic regression, expressed as odds ratios (O.R.; 95%CI; P-values). Standardised normal transformation of predictor variables was used in the regression models: ({respondent value minus group mean} /standard deviation). Estimates of the associations during each survey period, and the linear change per year, for each predictor were made using time (as a continuous variable) interaction terms in the regression models.

Where it was possible to identify repeated survey completions by the same respondent, this was done, although such identification was likely incomplete. Missing data for outcome and predictor variables were substituted by multiple imputation. 200 imputations were performed, using all available outcome and predictor variables, by the Multivariate Chained Monte Carlo method as implemented by Stata 16.1. Where the respondent gave a "Don't know" or "Prefer not to answer" or "Not applicable" response, this was treated as missing data. As a guide to data interpretation: an I.R.R. or O.R. of 1.00 indicates no rate change or no association, greater than 1.00 an increased rate or positive association, and less than 1.00 a decreased rate or negative association. All the listed predictors except Question 28 ("tangible workplace support" excluded due to low rate of usable responses) were included in the model for each separate outcome score.

Two sensitivity analyses were conducted, using the ordered logistic regression method described above: 1) to determine whether inclusion of all the variables affected the effect size estimates, compared to a more selective model inclusion policy; 2) to determine whether exclusion of 139 questionnaires due to apparent multiple responses from individual nurses in the same survey period changed the outcome estimates.

Details of the statistical analysis and the rationale for their use are described in Supplementary Materials. All quantitative analyses were performed using Stata MP2 version 16.1 (StataCorp LLC, College Station, Tx USA).

## Results

The participants in the survey were drawn from a defined sample of nurses and midwives by the nursing department email list of the Tasmanian Health Service (Fig 2). Being an anonymous survey there is no information about those who did not respond.

### Demographics

1499 of 1646 respondents (91%) were Registered Nurses (incorporating RN-Midwife), and 126 were Enrolled Nurses (8%) (see Table 1). 763 of 1645 respondents (46%) were over 50 years of age. The shows 857 of 1642 of participants (52%) had an educational level of graduate certificate or higher, and 511 (31%) held a bachelor's degree. 209 of 1631 respondents (13%) had looked after COVID-19 positive patients. Living situations remained consistent showing across all surveys with 82% of nurses and midwives living with family or friends and two percent living alone. Eight percent of nurses and midwives are smokers.

### Prevalence of psychological outcomes

Levels of anxiety, depression, insomnia and PTSD indicate significant psychological distress among respondents (see Table 2). Surveys 1, 2 and 3 show respectively 8%, 7% and 11% of

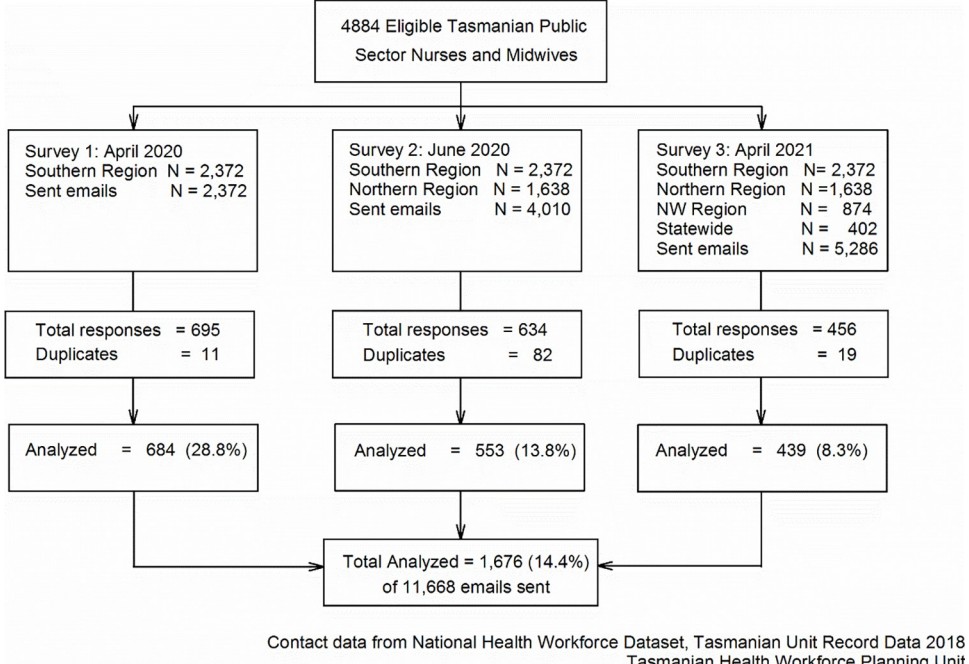

**Fig 2. Respondent flow diagram (National health workforce dataset, Tasmanian unit record data 2018).**

**Table 2. Prevalence of psychological distress in the surveys, and change in the rates of the distress over 1 year.**

| Psychological distress scale | Median (IQR) | n [1] | Rate [1] | 95%CI; ± P-value | n [1] | Rate [1] | 95%CI; ± P-value |
|---|---|---|---|---|---|---|---|
| **Anxiety (GAD-7)** | | GAD-7 ≥10 | | | GAD-7 ≥15 | | |
| Survey 1 (N = 684) | 5 (2, 9) | 154 | 22.5% | (19.4%, 25.8%) | 56 | 8.2% | (6.2%, 10.5%) |
| Survey 2 (N = 553) | 5 (1, 8) | 101 | 18.3% | (15.1%, 21.7%) | 41 | 7.4% | (5.4%, 9.9%) |
| Survey 3 (N = 439) | 5 (1, 8) | 90 | 20.5% | (16.8%, 24.6%) | 50 | 11.4% | (8.6%, 14.7%) |
| Change (IRR; 95%CI; P) [2] | | | 0.94 | (0.93, 1.17; P = 0.51) | | 1.42 | (1.00, 2.02; P = 0.053) |
| Depression (PHQ-9) | | PHQ-9 ≥10 | | | PHQ-9 ≥15 | | |
| Survey 1 (N = 684) | 5 (2, 9) | 177 | 25.9% | (22.6%, 29.3%) | 68 | 9.9% | (7.8%, 12.4%) |
| Survey 2 (N = 553) | 5 (2, 9) | 127 | 23.0% | (19.5%, 26.7%) | 66 | 11.9% | (9.4%, 14.9%) |
| Survey 3 (N = 439) | 5 (1, 10) | 121 | 27.6% | (23.4%, 32.0%) | 62 | 14.1% | (11.0%, 17.7%) |
| Change (IRR; 95%CI; P) [2] | | | 1.10 | (0.93, 1.32; P = 0.27) | | 1.39 | (1.07, 1.81; P = 0.015) |
| Insomnia (ISI) | | ISI ≥10 | | | ISI ≥15 | | |
| Survey 1 (N = 684) | 9 (5, 13) | 326 | 47.7% | (43.9%, 51.5%) | 131 | 19.2% | (16.3%, 22.3%) |
| Survey 2 (N = 553) | 9 (4, 13) | 248 | 19.2% | (16.0%, 22.7%) | 106 | 19.2% | (16.0%, 22.7%) |
| Survey 3 (N = 439) | 9 (5, 14) | 218 | 21.1% | (17.5%, 25.3%) | 93 | 21.1% | (17.5%, 25.3%) |
| Change (IRR; 95%CI; P) [2] | | | 1.06 | (0.95, 1.19; P = 0.30) | | 1.10 | (0.88, 1.37; P = 0.41) |
| Insomnia (Cont.) | | ISI ≥22 | | | | | |
| Survey 1 (N = 684) | | 14 | 2.0% | (1.1%, 3.4%) | | | |
| Survey 2 (N = 553) | | 20 | 3.6% | (2.2%, 5.5%) | | | |
| Survey 3 (N = 439) | | 19 | 4.3% | (2.6%, 6.7%) | | | |
| Change (IRR; 95%CI; P) [2] | | | 1.72 | (0.97, 3.05; P = 0.063) | | | |
| Stress, PTSD (IES-R) | | IES-R ≥33 | | | IES-R ≥37 | | |
| Survey 1 (N = 684) | 15 (6, 26) | 111 | 16.2% | (13.5%, 19.2%) | 76 | 11.1% | (8.9%, 13.7%) |
| Survey 2 (N = 553) | 12 (4, 23) | 64 | 11.6% | (9.0%, 14.5%) | 50 | 9.0% | (6.8%, 11.7%) |
| Survey 3 (N = 439) | 8 (2, 21) | 42 | 9.6% | (7.0%, 12.7%) | 34 | 7.7% | (5.4%, 10.7%) |
| Change (IRR; 95%CI; P) [2] | | | 0.59 | (0.42, 0.84; P = 0.0029) | | 0.69 | (0.46, 1.04; 0.078) |

[1] Rate of occurrence of cases above the stated distress scale threshold: n = case numbers; N = total response numbers; rate as percentage; exact binomial 95% confidence intervals; missing data substituted by multiple imputation

[2] Rate of change per year in occurrence of cases: Incidence rate ratio (I.R.R.; 95% confidence intervals; P-values) estimated by Poisson regression

respondents had severe anxiety (GAD-7 ≥10), 10%, 12% and 14% severe levels of depression (PHQ-9 ≥10), 2%, 4% and 4% severe levels of insomnia (ISI ≥22), and 11%, 9% and 8% with severe levels of PTSD (IES-R ≥37, high enough to cause immune suppression). Increasing levels of severe anxiety, depression, and insomnia, were seen over the study period, while severe PTSD-symptoms decrease. 112 of 1,676 (6.7%) respondents had moderate/severe levels of each of anxiety, depression, insomnia and PTSD-symptoms.

## Associations of psychological outcomes with possible predictors

The associations between each predictor variable and the four psychological distress outcomes, and how the association estimates changed over time, were estimated simultaneously by multiple regression analysis. The results are shown in Tables 3–6 with the predictors ordered to show those with the strongest positive associations first, and the strongest negative associations last. All the available predictors were included in the analysis in order that the relative position of each against the others could be appreciated.

Four associations were apparent from the analysis of the surveys, when the strength of the different predictors of psychological distress were compared against each other (see Tables 3 to 6). Firstly, when asked to identify the most important reasons for their level of psychological

**Table 3. Relative impact of predictors on Generalized Anxiety Disorder (GAD-7) score in responses over the year of surveys.**

| Predictors | Association between GAD-7 and predictor at each survey time period [1,2] | | | | | | | | | Change in association between GAD-7 and predictor over the year [1,2] | | |
|---|---|---|---|---|---|---|---|---|---|---|---|---|
| | Survey 1 (Initial; N = 684) | | | Survey 2 (3-month; N = 553) | | | Survey 3 (12-month; N = 439) | | | | | |
| | O.R. | 95%CI | P-value | O.R. | 95%CI | P-value | O.R. | 95%CI | P-value | O.R. | 95%CI | P-value |
| Change in GAD-7 over 1 year 3 | | | | | | | | | | 1.13 | (0.89, 1.44) | 0.31 |
| Individual predictors | | | | | | | | | | | | |
| Current home/family stress | 2.13 | (1.82, 2.50) | <0.001 | 1.97 | (1.72, 2.25) | <0.001 | 1.40 | (1.09, 1.80) | 0.008 | 0.66 | (0.49, 0.89) | 0.007 |
| Poor clinical team support | 1.27 | (1.11, 1.45) | 0.001 | 1.36 | (1.21, 1.53) | <0.001 | 1.82 | (1.47, 2.26) | <0.001 | 1.43 | (1.11, 1.84) | 0.005 |
| Future home/family stress | 1.18 | (1.01, 1.39) | 0.041 | 1.14 | (0.99, 1.30) | 0.063 | 0.97 | (0.74, 1.27) | 0.81 | 0.82 | (0.59, 1.13) | 0.23 |
| Inadequacy of information | 1.16 | (1.01, 1.33) | 0.039 | 1.14 | (1.01, 1.29) | 0.028 | 1.09 | (0.87, 1.37) | 0.46 | 0.94 | (0.72, 1.23) | 0.66 |
| Future financial stress | 1.16 | (0.98, 1.38) | 0.079 | 1.15 | (0.99, 1.32) | 0.062 | 1.08 | (0.77, 1.51) | 0.67 | 0.93 | (0.62, 1.38) | 0.70 |
| Poor ability for quality care | 1.13 | (0.98, 1.30) | 0.082 | 1.14 | (1.02, 1.29) | 0.025 | 1.21 | (0.96, 1.53) | 0.11 | 1.07 | (0.82, 1.41) | 0.62 |
| Covid ward status | 1.10 | (0.97, 1.25) | 0.13 | 1.07 | (0.96, 1.19) | 0.22 | 0.94 | (0.75, 1.18) | 0.58 | 0.85 | (0.65, 1.11) | 0.23 |
| Excessive information | 1.10 | (0.97, 1.24) | 0.13 | 1.09 | (0.98, 1.21) | 0.11 | 1.05 | (0.86, 1.27) | 0.65 | 0.95 | (0.76, 1.20) | 0.68 |
| Concerned about PPE | 1.10 | (0.96, 1.26) | 0.16 | 1.10 | (0.98, 1.24) | 0.097 | 1.10 | (0.89, 1.37) | 0.37 | 1.00 | (0.78, 1.30) | 0.98 |
| Smoking | 1.07 | (0.95, 1.20) | 0.28 | 1.08 | (0.97, 1.19) | 0.15 | 1.12 | (0.96, 1.31) | 0.15 | 1.05 | (0.86, 1.27) | 0.63 |
| Poor access to rapid tests | 1.04 | (0.92, 1.18) | 0.54 | 1.05 | (0.94, 1.17) | 0.39 | 1.08 | (0.88, 1.33) | 0.46 | 1.04 | (0.81, 1.33) | 0.76 |
| Enough deployment training | 1.04 | (0.91, 1.18) | 0.60 | 1.03 | (0.92, 1.16) | 0.57 | 1.02 | (0.83, 1.26) | 0.84 | 0.99 | (0.77, 1.26) | 0.91 |
| Public sector | 1.03 | (0.91, 1.17) | 0.63 | 1.03 | (0.92, 1.15) | 0.61 | 1.02 | (0.80, 1.30) | 0.89 | 0.99 | (0.74, 1.31) | 0.92 |
| Social situation | 1.03 | (0.90, 1.17) | 0.68 | 1.02 | (0.91, 1.13) | 0.76 | 0.97 | (0.80, 1.19) | 0.79 | 0.95 | (0.75, 1.21) | 0.66 |
| Lower staff grade | 0.99 | (0.87, 1.12) | 0.88 | 0.99 | (0.89, 1.11) | 0.88 | 1.00 | (0.80, 1.24) | 0.97 | 1.01 | (0.78, 1.31) | 0.96 |
| Current financial stress | 0.99 | (0.83, 1.18) | 0.89 | 1.04 | (0.89, 1.20) | 0.63 | 1.27 | (0.92, 1.76) | 0.15 | 1.29 | (0.88, 1.89) | 0.19 |
| Higher education level | 0.94 | (0.83, 1.06) | 0.31 | 0.92 | (0.83, 1.03) | 0.16 | 0.87 | (0.70, 1.07) | 0.18 | 0.93 | (0.73, 1.18) | 0.54 |
| Intensity of exposure | 0.91 | (0.79, 1.05) | 0.19 | 0.95 | (0.84, 1.07) | 0.37 | 1.12 | (0.91, 1.38) | 0.30 | 1.22 | (0.95, 1.58) | 0.12 |
| Males | 0.85 | (0.76, 0.96) | 0.007 | 0.85 | (0.77, 0.94) | 0.002 | 0.84 | (0.68, 1.02) | 0.083 | 0.98 | (0.78, 1.24) | 0.88 |
| Age group | 0.75 | (0.66, 0.85) | <0.001 | 0.72 | (0.64, 0.80) | <0.001 | 0.61 | (0.49, 0.76) | <0.001 | 0.82 | (0.64, 1.05) | 0.11 |

[1] Associations between predictors and outcomes, and change in outcomes over 1 year were estimated using repeated-measures ordered logistic regression; the effects were shown as odds ratios (O.R.; 95% confidence intervals; P-values), adjusted for each of the predictors shown in the tables; missing data was substituted by multiple imputation. Odds ratios of 1.00 indicate no association; O.R. more than 1.00 indicate a positive association; O.R. less than 1.00 indicate a negative association.

[2] The association between predictors and outcomes in the surveys were estimated for the mean survey completion dates for each survey, set as the zero value of the time interaction predictor: (date of each survey completion by specific respondent minus the mean date of each survey completion). Mean date of survey completion was: Survey 1, 0.1 years on 6/05/2020; Survey 2, 0.29 years on 13/07/2020; Survey 3, 1.1 years 8/05/2021. The change in association between each predictor and outcomes was determined by the time interaction.

[3] Model fit parameters: Log likelihood (null) -4675.94; Log likelihood (model) -4320.87; df 62; Akaike information criterion (AIC) 8765.744; Bayesian information criterion (BIC) 9102.042

distress, in the initial survey, current home and family stress was nominated as the strongest or most common reason for each of all of their anxiety-, insomnia-, PTSD-, and depression-related symptoms. In the second and third surveys, the strength of this association weakened compared to the other predictors, but it remained in each case the second most important association as judged by the nurses and midwives. Future home and family stress was regarded as less distressful as current, and its associated diminished in parallel with that of current home and family stress, as displayed in Figs 3–5.

Secondly, poor clinical team support was one of the more predominant associations at the initial survey. The strength of this association increased for each of the four psychological domains, and by the third survey, it was judged by the nurses and midwives as being the strongest association. It was one of three predictors that showed consistent increase over the year of the surveys. This was true whether the overall distress remained the same (for anxiety),

**Table 4. Relative impact of predictors on Insomnia Severity Index (ISI) score in responses over the year of surveys.**

| Predictors | Association between ISI and predictor at each survey time period [1,2] | | | | | | | | | Change in association between ISI and predictor over the year [1,2] | | |
| | Survey 1 (Initial; N = 684) | | | Survey 2 (3-month; N = 553) | | | Survey 3 (12-month; N = 439) | | | | | |
| | O.R. | 95%CI | P-value | O.R. | 95%CI | P-value | O.R. | 95%CI | P-value | O.R. | 95%CI | P-value |
|---|---|---|---|---|---|---|---|---|---|---|---|---|
| Change in ISI over 1 year 3 | | | | | | | | | | 1.41 | (1.13, 1.77) | 0.003 |
| Individual predictors | | | | | | | | | | | | |
| Current home/family stress | 1.63 | (1.38, 1.92) | <0.001 | 1.57 | (1.37, 1.81) | <0.001 | 1.34 | (1.04, 1.73) | 0.024 | 0.82 | (0.60, 1.13) | 0.23 |
| Future home/family stress | 1.26 | (1.07, 1.49) | 0.007 | 1.18 | (1.03, 1.36) | 0.020 | 0.89 | (0.70, 1.14) | 0.35 | 0.71 | (0.52, 0.96) | 0.025 |
| Poor clinical team support | 1.20 | (1.04, 1.37) | 0.011 | 1.25 | (1.11, 1.40) | <0.001 | 1.51 | (1.23, 1.85) | <0.001 | 1.26 | (0.98, 1.61) | 0.071 |
| Poor access to rapid tests | 1.17 | (1.02, 1.34) | 0.022 | 1.17 | (1.05, 1.31) | 0.006 | 1.18 | (0.97, 1.45) | 0.097 | 1.01 | (0.79, 1.30) | 0.92 |
| Concerned about PPE | 1.13 | (0.99, 1.29) | 0.080 | 1.13 | (1.01, 1.27) | 0.036 | 1.14 | (0.94, 1.39) | 0.19 | 1.01 | (0.80, 1.29) | 0.92 |
| Current financial stress | 1.13 | (0.95, 1.35) | 0.18 | 1.18 | (1.01, 1.37) | 0.035 | 1.39 | (1.06, 1.82) | 0.017 | 1.23 | (0.88, 1.71) | 0.23 |
| Poor ability for quality care | 1.12 | (0.97, 1.28) | 0.11 | 1.10 | (0.98, 1.24) | 0.097 | 1.05 | (0.82, 1.35) | 0.69 | 0.94 | (0.71, 1.26) | 0.70 |
| Inadequacy of information | 1.11 | (0.96, 1.30) | 0.16 | 1.10 | (0.97, 1.25) | 0.14 | 1.05 | (0.83, 1.32) | 0.67 | 0.94 | (0.71, 1.25) | 0.69 |
| Covid ward status | 1.10 | (0.96, 1.26) | 0.18 | 1.07 | (0.95, 1.21) | 0.28 | 0.96 | (0.75, 1.22) | 0.72 | 0.87 | (0.66, 1.15) | 0.33 |
| Enough deployment training | 1.10 | (0.95, 1.27) | 0.19 | 1.09 | (0.97, 1.23) | 0.16 | 1.06 | (0.85, 1.31) | 0.61 | 0.96 | (0.74, 1.25) | 0.77 |
| Future financial stress | 1.04 | (0.87, 1.25) | 0.67 | 1.06 | (0.91, 1.23) | 0.46 | 1.14 | (0.86, 1.51) | 0.35 | 1.10 | (0.78, 1.55) | 0.60 |
| Excessive information | 1.03 | (0.92, 1.16) | 0.61 | 1.05 | (0.95, 1.16) | 0.30 | 1.16 | (0.94, 1.42) | 0.18 | 1.12 | (0.88, 1.43) | 0.37 |
| Public sector | 1.03 | (0.90, 1.18) | 0.66 | 1.06 | (0.95, 1.20) | 0.30 | 1.21 | (0.98, 1.49) | 0.071 | 1.17 | (0.91, 1.50) | 0.21 |
| Social situation | 1.01 | (0.88, 1.16) | 0.88 | 1.02 | (0.91, 1.14) | 0.79 | 1.03 | (0.88, 1.22) | 0.68 | 1.02 | (0.83, 1.27) | 0.83 |
| Lower staff grade | 1.00 | (0.89, 1.13) | 0.96 | 0.99 | (0.89, 1.09) | 0.77 | 0.91 | (0.76, 1.10) | 0.34 | 0.91 | (0.73, 1.14) | 0.42 |
| Males | 1.00 | (0.88, 1.12) | 0.94 | 0.96 | (0.86, 1.07) | 0.46 | 0.83 | (0.67, 1.02) | 0.069 | 0.83 | (0.66, 1.05) | 0.12 |
| Smoking | 0.99 | (0.85, 1.14) | 0.84 | 1.01 | (0.89, 1.14) | 0.85 | 1.13 | (0.97, 1.32) | 0.11 | 1.15 | (0.94, 1.40) | 0.17 |
| Age group | 0.98 | (0.87, 1.11) | 0.74 | 0.96 | (0.86, 1.07) | 0.46 | 0.88 | (0.71, 1.09) | 0.24 | 0.90 | (0.70, 1.15) | 0.40 |
| Intensity of exposure | 0.95 | (0.82, 1.11) | 0.54 | 0.99 | (0.87, 1.12) | 0.85 | 1.15 | (0.94, 1.40) | 0.18 | 1.20 | (0.93, 1.55) | 0.16 |
| Higher education level | 0.76 | (0.67, 0.87) | <0.001 | 0.77 | (0.68, 0.86) | <0.001 | 0.79 | (0.65, 0.97) | 0.022 | 1.04 | (0.82, 1.31) | 0.76 |

[1] Associations between predictors and outcomes, and change in outcomes over 1 year were estimated using repeated-measures ordered logistic regression; the effects were shown as odds ratios (O.R.; 95% confidence intervals; P-values), adjusted for each of the predictors shown in the tables; missing data was substituted by multiple imputation. Odds ratios of 1.00 indicate no association; O.R. more than 1.00 indicate a positive association; O.R. less than 1.00 indicate a negative association.

[2] The association between predictors and outcomes in the surveys were estimated for the mean survey completion dates for each survey, set as the zero value of the time interaction predictor: (date of each survey completion by specific respondent minus the mean date of each survey completion). Mean date of survey completion was: Survey 1, 0.1 years on 6/05/2020; Survey 2, 0.29 years on 13/07/2020; Survey 3, 1.1 years 8/05/2021. The change in association between each predictor and outcomes was determined by the time interaction.

[3] Model fit parameters: Log likelihood (null) -5205.15; Log likelihood (model) -4945.69; df 69; Akaike information criterion (AIC) 10029.4; Bayesian information criterion (BIC) 10403.64

decreased (for PTSD) or increased (for insomnia and depression), although the details of the increase varied (Fig 4). The other two predictors that increased in prominence were current financial stress especially insomnia and PTSD symptoms, and intensity of exposure to COVID-19, although these two predictors were not prominent in the initial survey.

Thirdly, older nurses and midwives were less anxious and depressed than younger colleagues. There was no tendency for this difference to reduce over the year. Fourthly, males had less anxiety, depression and PTSD symptomatology, and again there was no tendency for these differences to change.

The analysis allowed comparison of what was important, and what was of lesser importance. There were a number of predictors that showed mild association with psychological distress. These were either mild associations, or stronger associations for only a minority of respondents (the analyses could not distinguish between these possibilities). These included

**Table 5. Relative impact of predictors on Impact of Events–Revised (IES-R) score in responses over the year of surveys.**

| | Association between IES-R and predictor at each survey time period [1,2] | | | | | | | | | Change in association between IES-R and predictor over the year [1,2] | | |
| | Survey 1 (Initial; N = 684) | | | Survey 2 (3-month; N = 553) | | | Survey 3 (12-month; N = 439) | | | | | |
| Predictors | O.R. | 95%CI | P-value | O.R. | 95%CI | P-value | O.R. | 95%CI | P-value | O.R. | 95%CI | P-value |
|---|---|---|---|---|---|---|---|---|---|---|---|---|
| **Change in IES-R over 1 year 3** | | | | | | | | | | **0.64** | **(0.51, 0.81)** | **<0.001** |
| Individual predictors | | | | | | | | | | | | |
| Current home/family stress | 1.93 | (1.61, 2.30) | <0.001 | 1.85 | (1.60, 2.14) | <0.001 | 1.56 | (1.22, 1.98) | <0.001 | 0.81 | (0.59, 1.10) | 0.18 |
| Poor clinical team support | 1.30 | (1.13, 1.50) | <0.001 | 1.36 | (1.20, 1.54) | <0.001 | 1.62 | (1.29, 2.03) | <0.001 | 1.24 | (0.95, 1.62) | 0.12 |
| Concerned about PPE | 1.29 | (1.13, 1.47) | <0.001 | 1.28 | (1.14, 1.42) | <0.001 | 1.21 | (0.99, 1.48) | 0.058 | 0.94 | (0.73, 1.20) | 0.61 |
| Future home/family stress | 1.23 | (1.04, 1.47) | 0.017 | 1.22 | (1.06, 1.40) | 0.007 | 1.14 | (0.87, 1.50) | 0.33 | 0.93 | (0.66, 1.30) | 0.67 |
| Poor ability for quality care | 1.21 | (1.05, 1.39) | 0.006 | 1.17 | (1.04, 1.32) | 0.008 | 1.02 | (0.80, 1.30) | 0.87 | 0.85 | (0.63, 1.13) | 0.25 |
| Inadequacy of information | 1.18 | (1.03, 1.35) | 0.015 | 1.18 | (1.06, 1.32) | 0.004 | 1.19 | (0.95, 1.48) | 0.13 | 1.01 | (0.77, 1.31) | 0.97 |
| Future financial stress | 1.18 | (0.99, 1.42) | 0.066 | 1.18 | (1.01, 1.37) | 0.033 | 1.16 | (0.85, 1.60) | 0.34 | 0.98 | (0.68, 1.43) | 0.93 |
| Excessive information | 1.12 | (0.99, 1.26) | 0.074 | 1.11 | (1.00, 1.23) | 0.044 | 1.09 | (0.90, 1.32) | 0.37 | 0.98 | (0.78, 1.23) | 0.85 |
| Social situation | 1.11 | (0.98, 1.27) | 0.11 | 1.09 | (0.98, 1.22) | 0.12 | 1.00 | (0.83, 1.21) | 0.98 | 0.90 | (0.72, 1.14) | 0.38 |
| Covid ward status | 1.07 | (0.94, 1.22) | 0.29 | 1.06 | (0.95, 1.20) | 0.30 | 1.02 | (0.79, 1.32) | 0.86 | 0.95 | (0.72, 1.26) | 0.74 |
| Current financial stress | 1.07 | (0.90, 1.28) | 0.46 | 1.13 | (0.97, 1.31) | 0.12 | 1.41 | (1.05, 1.90) | 0.022 | 1.32 | (0.93, 1.87) | 0.12 |
| Smoking | 1.04 | (0.93, 1.17) | 0.48 | 1.07 | (0.97, 1.18) | 0.18 | 1.20 | (0.97, 1.48) | 0.095 | 1.15 | (0.90, 1.47) | 0.28 |
| Intensity of exposure | 1.04 | (0.90, 1.19) | 0.59 | 1.07 | (0.95, 1.20) | 0.28 | 1.19 | (0.97, 1.47) | 0.087 | 1.15 | (0.89, 1.48) | 0.28 |
| Public sector | 1.03 | (0.89, 1.20) | 0.66 | 1.04 | (0.92, 1.18) | 0.56 | 1.05 | (0.89, 1.25) | 0.54 | 1.02 | (0.81, 1.28) | 0.87 |
| Enough deployment training | 1.02 | (0.89, 1.17) | 0.78 | 1.04 | (0.93, 1.17) | 0.46 | 1.16 | (0.94, 1.42) | 0.17 | 1.13 | (0.89, 1.45) | 0.32 |
| Age group | 0.95 | (0.83, 1.09) | 0.49 | 0.95 | (0.85, 1.07) | 0.42 | 0.95 | (0.77, 1.18) | 0.66 | 1.00 | (0.78, 1.29) | 1.00 |
| Poor access to rapid tests | 0.95 | (0.83, 1.08) | 0.45 | 0.96 | (0.86, 1.07) | 0.43 | 0.98 | (0.79, 1.22) | 0.87 | 1.03 | (0.79, 1.35) | 0.82 |
| Lower staff grade | 0.90 | (0.79, 1.02) | 0.099 | 0.90 | (0.80, 1.00) | 0.044 | 0.89 | (0.73, 1.07) | 0.21 | 0.99 | (0.78, 1.25) | 0.91 |
| Higher education level | 0.88 | (0.78, 1.00) | 0.056 | 0.87 | (0.78, 0.97) | 0.014 | 0.83 | (0.68, 1.00) | 0.050 | 0.93 | (0.74, 1.17) | 0.56 |
| Males | 0.80 | (0.71, 0.90) | <0.001 | 0.81 | (0.73, 0.90) | <0.001 | 0.85 | (0.70, 1.04) | 0.13 | 1.07 | (0.84, 1.36) | 0.58 |

[1] Associations between predictors and outcomes, and change in outcomes over 1 year were estimated using repeated-measures ordered logistic regression; the effects were shown as odds ratios (O.R.; 95% confidence intervals; P-values), adjusted for each of the predictors shown in the tables; missing data was substituted by multiple imputation. Odds ratios of 1.00 indicate no association; O.R. more than 1.00 indicate a positive association; O.R. less than 1.00 indicate a negative association.

[2] The association between predictors and outcomes in the surveys were estimated for the mean survey completion dates for each survey, set as the zero value of the time interaction predictor: (date of each survey completion by specific respondent minus the mean date of each survey completion). Mean date of survey completion was: Survey 1, 0.1 years on 6/05/2020; Survey 2, 0.29 years on 13/07/2020; Survey 3, 1.1 years 8/05/2021. The change in association between each predictor and outcomes was determined by the time interaction.

[3] Model fit parameters: Log likelihood (null) -6229.95; Log likelihood (model) -5812.02; df 113; Akaike information criterion (AIC) 11850.03; Bayesian information criterion (BIC) 12462.96

excess and inadequate information related to COVID-19, access to rapid COVID-19 testing, ability to provide quality patient care, and access to PPE.

***Post-hoc* sensitivity analyses.** Firstly, to determine whether the inclusion of all available predictors produced different effects compared to a more selective inclusion of predictors. This analysis showed no meaningful alteration depending on the inclusion or exclusion of low-association variables (see Tables A7 to A10 in S3 File). Secondly, 139 completed questionnaires that were judged to be duplicates (the same person completing the survey more than once per survey period) and excluded from the main analysis. Sensitivity analysis showed no meaningful alteration in results found when all 1,815 responses received were compared to the 1,676 responses in the main analysis after the exclusion of the apparent duplicates (see Tables A11 to A14 in S1 File).

**Table 6. Relative impact of predictors on Patient Health Questionnaire (PHQ-9) score in responses over the year of surveys.**

| | Association between PHQ-9 and predictor at each survey time period [1,2] | | | | | | | | | Change in association between PHQ-9 and predictor over the year [1,2] | | |
| | Survey 1 (Initial; N = 684) | | | Survey 2 (3-month; N = 553) | | | Survey 3 (12-month; N = 439) | | | | | |
| Predictors | O.R. | 95%CI | P-value | O.R. | 95%CI | P-value | O.R. | 95%CI | P-value | O.R. | 95%CI | P-value |
|---|---|---|---|---|---|---|---|---|---|---|---|---|
| **Change in PHQ-9 over 1 year 3** | | | | | | | | | | 1.22 | (0.95, 1.57) | 0.12 |
| Individual predictors | | | | | | | | | | | | |
| Current home/family stress | 1.65 | (1.38, 1.96) | <0.001 | 1.58 | (1.37, 1.83) | <0.001 | 1.34 | (1.04, 1.72) | 0.024 | 0.81 | (0.59, 1.11) | 0.20 |
| Future home/family stress | 1.25 | (1.05, 1.48) | 0.010 | 1.20 | (1.04, 1.39) | 0.011 | 1.02 | (0.78, 1.32) | 0.91 | 0.81 | (0.58, 1.13) | 0.22 |
| Poor ability for quality care | 1.20 | (1.04, 1.39) | 0.012 | 1.18 | (1.05, 1.34) | 0.007 | 1.11 | (0.87, 1.42) | 0.40 | 0.92 | (0.69, 1.23) | 0.59 |
| Social situation | 1.19 | (1.04, 1.36) | 0.012 | 1.18 | (1.05, 1.32) | 0.005 | 1.13 | (0.92, 1.39) | 0.25 | 0.95 | (0.74, 1.22) | 0.68 |
| Poor clinical team support | 1.18 | (1.02, 1.36) | 0.027 | 1.26 | (1.12, 1.43) | <0.001 | 1.72 | (1.39, 2.12) | <0.001 | 1.46 | (1.12, 1.89) | 0.005 |
| Inadequacy of information | 1.17 | (1.00, 1.35) | 0.047 | 1.17 | (1.03, 1.33) | 0.016 | 1.19 | (0.93, 1.51) | 0.16 | 1.02 | (0.76, 1.36) | 0.90 |
| Enough deployment training | 1.11 | (0.96, 1.28) | 0.17 | 1.11 | (0.98, 1.26) | 0.091 | 1.13 | (0.93, 1.38) | 0.22 | 1.02 | (0.80, 1.31) | 0.87 |
| Excessive information | 1.10 | (0.97, 1.25) | 0.15 | 1.10 | (0.98, 1.22) | 0.097 | 1.09 | (0.90, 1.32) | 0.40 | 0.99 | (0.78, 1.25) | 0.93 |
| Smoking | 1.09 | (0.96, 1.23) | 0.17 | 1.10 | (0.99, 1.22) | 0.071 | 1.15 | (0.95, 1.39) | 0.15 | 1.06 | (0.84, 1.32) | 0.64 |
| Future financial stress | 1.09 | (0.91, 1.30) | 0.37 | 1.10 | (0.94, 1.27) | 0.23 | 1.14 | (0.82, 1.58) | 0.43 | 1.05 | (0.70, 1.56) | 0.82 |
| Concerned about PPE | 1.08 | (0.94, 1.24) | 0.29 | 1.10 | (0.97, 1.23) | 0.13 | 1.18 | (0.96, 1.46) | 0.12 | 1.09 | (0.85, 1.41) | 0.49 |
| Current financial stress | 1.07 | (0.89, 1.28) | 0.46 | 1.08 | (0.93, 1.26) | 0.30 | 1.14 | (0.84, 1.55) | 0.40 | 1.06 | (0.74, 1.53) | 0.74 |
| Covid ward status | 1.03 | (0.90, 1.19) | 0.65 | 1.02 | (0.90, 1.15) | 0.78 | 0.95 | (0.74, 1.23) | 0.72 | 0.92 | (0.69, 1.24) | 0.60 |
| Intensity of exposure | 1.03 | (0.89, 1.18) | 0.74 | 1.04 | (0.92, 1.18) | 0.50 | 1.12 | (0.90, 1.39) | 0.31 | 1.09 | (0.83, 1.43) | 0.52 |
| Public sector | 1.01 | (0.87, 1.17) | 0.91 | 0.99 | (0.87, 1.12) | 0.85 | 0.90 | (0.72, 1.12) | 0.36 | 0.89 | (0.69, 1.16) | 0.40 |
| Higher education level | 0.98 | (0.85, 1.12) | 0.72 | 0.95 | (0.85, 1.07) | 0.42 | 0.87 | (0.71, 1.05) | 0.15 | 0.89 | (0.70, 1.13) | 0.34 |
| Poor access to rapid tests | 0.97 | (0.84, 1.12) | 0.67 | 0.98 | (0.87, 1.10) | 0.69 | 1.01 | (0.81, 1.24) | 0.96 | 1.04 | (0.80, 1.35) | 0.79 |
| Lower staff grade | 0.94 | (0.83, 1.07) | 0.37 | 0.96 | (0.86, 1.07) | 0.43 | 1.02 | (0.84, 1.25) | 0.82 | 1.08 | (0.85, 1.38) | 0.52 |
| Males | 0.89 | (0.78, 1.01) | 0.061 | 0.88 | (0.79, 0.98) | 0.019 | 0.85 | (0.68, 1.06) | 0.14 | 0.96 | (0.74, 1.24) | 0.74 |
| Age group | 0.78 | (0.68, 0.89) | <0.001 | 0.76 | (0.68, 0.86) | <0.001 | 0.70 | (0.57, 0.86) | 0.001 | 0.90 | (0.70, 1.16) | 0.42 |

[1] Associations between predictors and outcomes, and change in outcomes over 1 year were estimated using repeated-measures ordered logistic regression; the effects were shown as odds ratios (O.R.; 95% confidence intervals; P-values), adjusted for each of the predictors shown in the tables; missing data was substituted by multiple imputation. Odds ratios of 1.00 indicate no association; O.R. more than 1.00 indicate a positive association; O.R. less than 1.00 indicate a negative association.

[2] The association between predictors and outcomes in the surveys were estimated for the mean survey completion dates for each survey, set as the zero value of the time interaction predictor: (date of each survey completion by specific respondent minus the mean date of each survey completion). Mean date of survey completion was: Survey 1, 0.1 years on 6/05/2020; Survey 2, 0.29 years on 13/07/2020; Survey 3, 1.1 years 8/05/2021. The change in association between each predictor and outcomes was determined by the time interaction.

[3] Model fit parameters: Log likelihood (null) -5010.69; Log likelihood (model) -4692.37; df 68; Akaike information criterion (AIC) 9520.73; Bayesian information criterion (BIC) 9889.58

## Thematic analysis

Nurses and midwives discussed their perceptions, experiences and memories of their COVID-19 experience at each survey period. Narratives are subjective and may be influenced and altered with time. Narratives contained both factual and emotional memories, some explicit often chronologically pieced together. Narrations were broadly separated into five themes and associated subthemes.

**Theme 1: Miscommunication—Subtheme: Inadequate leadership.** Narratives uncovered nurse and midwives' frustration around workplace miscommunication citing excessive communications yet receiving inadequate information. The lack of clear communication led to inconsistent advice and planning and confusion throughout the health service. With the lack of clear communication from the workplace, many nurses turned to alternative sources of

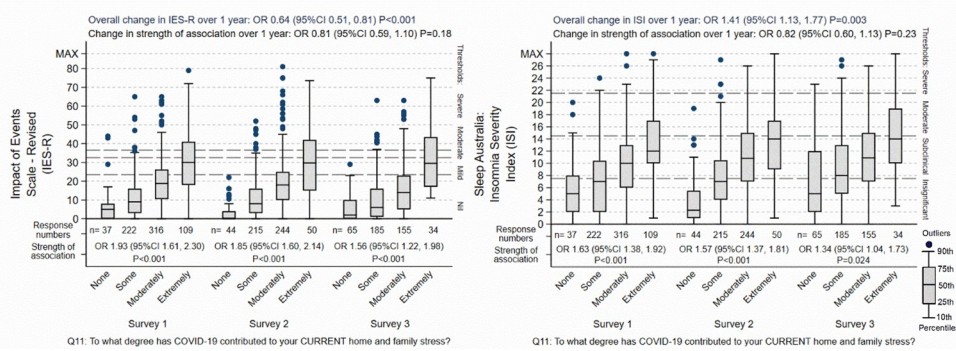

**Fig 3. Effect of home / family stress on IES-R and ISI.** The "Overall change over 1 year" is the background outcome scale O.R. for change in the respondents not explained by the predictor variable included in the regression models. The "Change in strength of association" is the O.R. for the change of the association of the specific predictor shown in the graph over 1 year, adjusted for the other predictors in the regression models. The box-plots display the group score distributions without adjustment for other predictors. The number of specific responses to the questions are shown at each response and survey period. The "Strength of association" is the association of between the outcome and predictor at each survey period.

information such as the general and social media. Professional and personal communications became blurred leaving staff feeling unsupported and perpetuated the spread of inaccurate information. The overall strategy of frontline nurses and midwives showed lack of organisation resulting in confusion: "It was hard to get or discuss any clear information each day. The NUMs [Nurse Unit Managers] stopped having ward meetings. We knew things were changing constantly but had no idea what we should be doing. We all just did whatever we thought was a reasonable thing" (T1312). There was confusion around the use of evolving COVID-19 terminology with classifications of hot, cold, red, orange zones and their associated levels of actioned response. Rapidly changing directions along with inconsistent infection control processes including access to supplies across different clinical areas compounded this confusion. Nurses and midwives spoke of much variation around training and use of PPE, expressing ethical stress related to having to adopt practices which put themselves and patients at risk: "The process of patient handover became disjoint and dangerous." (T264). There was frustration

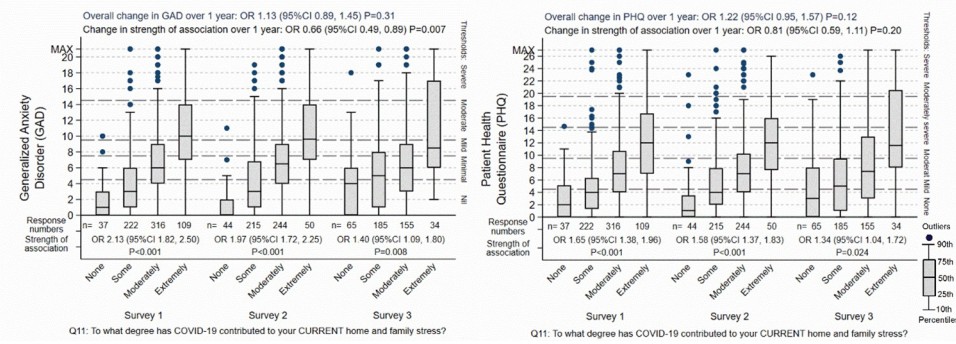

**Fig 4. Effect of home / family stress on GAD-7 and PHQ-9.** The "Overall change over 1 year" is the background outcome scale O.R. for change in the respondents not explained by the predictor variable included in the regression models. The "Change in strength of association" is the O.R. for the change of the association of the specific predictor shown in the graph over 1 year, adjusted for the other predictors in the regression models. The box-plots display the group score distributions without adjustment for other predictors. The number of specific responses to the questions are shown at each response and survey period. The "Strength of association" is the association of between the outcome and predictor at each survey period.

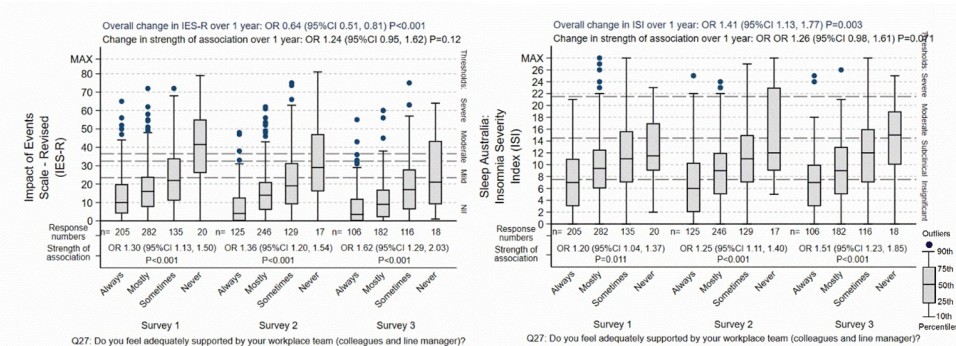

**Fig 5. Effect of poor clinical team support on IES-R and ISI.** The "Overall change over 1 year" is the background outcome scale O.R. for change in the respondents not explained by the predictor variable included in the regression models. The "Change in strength of association" is the O.R. for the change of the association of the specific predictor shown in the graph over 1 year, adjusted for the other predictors in the regression models. The box-plots display the group score distributions without adjustment for other predictors. The number of specific responses to the questions are shown at each response and survey period. The "Strength of association" is the association of between the outcome and predictor at each survey period.

and confusion related to changes to hospital infrastructure and associated models of care, staff leave entitlements including pandemic leave and the perceived lack of organisational support, all of which were changing daily. Initially social distancing was not a priority though slowly staff were supported to change work practices to reduce density of staff and patients in clinical areas.

There was much role change with new administrative or managerial positions created as part of the COVID-19 taskforce. Adopting new roles during an emergency, developing new lines of authority and accountability without adequate training provided many unique challenges. Nurses and midwives expressed anger and disappointment: "The greatest stress for me is feeling unsupported by management and their refusal to acknowledge their appalling communication" (T1238).

Overall nurses and midwives felt there was a lack of acknowledgement and understanding of the impact of COVID-19 on their role change by workplace management. Nurses and midwives described a department culture of yelling, belittling, passive aggression resulting in low job satisfaction. Nurses and midwives described tense relationships with management with little recognition of staff: ". . .we are always pushed to do more and more; management does not acknowledge how tired and stressed staff are" (T2314). Nurses and midwives attempted to organise co-worker support: "I think I would have liked regular debriefing and problem-solving sessions on a ward level. . ..we eventually set up a [private social platform] group, but . . .our temporary manager at the time did not like us venting our frustrations. There was an online stress reduction program but finding time to do that after work was not realistic and found it did not address specific concerns I had. Mostly about how do I manage my home life and be a nurse at the same time and keep everyone safe. . . ." (T3269).

Respondents described feeling unprepared and disjointed in the initial months, with a lack of organised service planning causing nurses and midwives to feel confused, frightened, uncertain, and unsupported. There was growing concern and uncertainty about COVID-19 transmission; they needed advice regarding work, the risk to family, and the potential harm they posed as a viral vector.

**Theme 2: Occupational risk—Subthemes: Vulnerability, devalued, blurred roles.**
Nurses and midwives felt ill-prepared and vulnerable as frontline health care workers. This stemmed from inadequate training, inadequate staff support protocols, insufficient PPE

supplies and access, lack of faith in the organisations' infection control procedures, comorbid physical health conditions and older age. The organisation attempted to meet a need for consistent PPE training by providing a PPE training video, which became both a support and a stressor. While its availability on demand was supportive, its inadequacy as a training method became a stressor, especially those for whom PPE use was unfamiliar, ". . . there was no PPE [allocated] for training. We don't use PPE day-to-day, so it was all new for us. We were told to watch a video" (T2374). The introduction of PPE buddies exposed knowledge and practice deficits among staff who regularly use PPE, leaving staff feeling scrutinised. Working in PPE is hot, physically restrictive, makes communication difficult and is both physically and mentally exhausting [37]. Many respondents felt vulnerable, devalued with a lack of professional and personal regard. Nurses and midwives felt they were fighting a losing battle, attempting to altruistically create boundaries of defence between work, family and society: "my ward has no changing facilities or showering facilities and minimal staff facilities. All staff are expected to change in a toilet cubicle (we have 1 toilet) and there's only a sink in the cubicle where they can clean themselves after the shift. I'm informed that senior staff have discussed with the ward manager the lack of facilities and they were basically told it is what it is" (T3384). Responses included managing family welfare with the welfare of their patients, as their professional and personal boundaries began to blur. Nurses and midwives with physical comorbidities or were immunocompromised, deemed high-risk, and those with young children or elderly family felt vulnerable, with many immediately withdrawing from work to reduce the risk of potential transmission. They were very aware of the consequences of their actions: "I have immunosuppressed people in my immediate family so I am not able to see my family. I am also petrified of the media and how they would villainise me if I would become positive as they have done to my colleagues" (T2188). Many older nurses and midwives either came to work, feeling scared, vulnerable, having to take on extremely busy workloads or withdrew from the workforce, precipitating many to retire. This resulted in a sudden loss of experienced nurses and midwives from the workforce. Nurses and midwives felt disappointed and appalled many pre-existing occupational health risks they faced were not acknowledged but put aside, to only focus on COVID-19 ". . ... [I am] concerned about the level of risk we are put under every day at work. What if I catch a superbug, get abused, night shifts and body clock/diet dysfunction, other influenza-like illnesses while we only seem to concentrate on COVID19. It been a hard few months" ID (T382). Attempts to support nurses and midwives were seen as token gestures and did not target their needs: "I think that this approach of 'oh we'll send out a brochure or tell people about the EAP [Employee Assistance Program] or what have you is totally missing the point. Healthcare workers need practical assistance with the difficulties caused by this crisis" (T381). Just like the general population, nurse and midwives' social lifestyles were impacted by the pandemic with gym closures and limited social opportunities which impacted their ability to cope with the ever-changing and threatening environment. Nurses and midwives acknowledge their work environment is normally stressful [38, 39] and develop strategies to manage this professional stress which were now inaccessible.

Responses included suffering from pre-existing mental health issues found these were compounded: "in the last two weeks I have had a depressive episode related to my mental health illness not related to the COVID pandemic, but the COVID pandemic has added another stressor to the mix" (T251). Negative coping mechanisms such as substance misuse, increased alcohol and over-eating were evident creating further psychological distress and precipitated mental health disorders. Many nurses and midwives replaced exercise with alcohol, regarding this as benign: "A glass or two of wine have become a daily habit in my house just to take the edge off" (T2473). Maintenance of their own personal health proved difficult. Nurses and midwives also endured the difficulties accessing medical and health care treatment, moving to

telehealth for their own healthcare. There were many comments of fear and stress as nurses waited for COVID-19 test results; fearful of media portrayal as a COVID-19 positive health care worker. Nurses and midwives were aware their workplace decisions and actions could potentially result in nosocomial spread, and front-page news, yet guidelines and protocols upon which they based these decisions did not exist. To society, nurses and midwives were both heroes and villains. Nurses and midwives endured societal stigma, being regarded as COVID-19 vectors. They felt vulnerable travelling to and from work in uniform, many enduring public verbal attacks and were angered by the changeable public perceptions, "I am annoyed that today I am regarded as a hero yet I am just doing the same job I did yesterday" (T171). From being valued and respected professionals, nurses and midwives were suddenly regarded as villains and shunned.

The vulnerability of patients caused much stress and anxiety for nurses and midwives and were sensitive to the disruption of patient treatments and services understanding the difficulty many patients had accessing routine health care. Change in practice, coping with staff shortages and less face-to-face time with patients caused much general and ethical stress for nurses and midwives, feeling their ability to provide quality patient care was compromised. Through the nature of their role, nurses and midwives hold an intimate, 'insider' understanding of the healthcare system and the barriers to acceptable access and treatment. They experienced stress associated with blurring of their role when loved ones were denied or subject to unacceptable treatment: "my father is having weekly radiotherapy and chemotherapy which has been missed because different staff give him different instructions" (T3203).

**Theme 3: Disconnect–Subthemes: Permeable boundaries, lack of social supports, economic uncertainty.** As time progressed together with the low COVID-19 case numbers seen in Tasmania, nurses and midwives were finding the impact of COVID-19 less, compared with the impact of other personal stresses. Nurses and midwives shared details of home and family stress: "it is difficult to separate feelings and life impacts of COVID and other personal circumstances when they all happened at the same time. . . .was a tumultuous time in my personal life including divorce, homelessness and job stress. (T312).

With the work environment rapidly changing, so was the nurses' social environment. Like the rest of the community, nurses and midwives faced personal lifestyle struggles. These included home schooling, caring for children with disabilities, managing children with dual custody, providing ongoing support to elderly relatives, the impact of family unemployment and the change to online tertiary education. Comments uncovered domestic violence. Tensions between partners were seen to increase as daily routines changed, roles changed, personal space at home was reduced as nurses and midwives (and their partners) worked from home, and financial pressures increased. Relationships were described as unhappy, yelling, frightened and scared, bullied, unpredictable, controlling, drinking a lot. Nurses and midwives disclosed intimate partner violence: ". . .we already have relationship problems and now with significantly less personal space I feel scared and stressed" (T1327). Comments included the value of clinical team support derived from their co-workers being one of the positive attributes of going to work.

As COVID-19 related stressors increased, nurses and midwives described an increase in tensions, fatigue and loss of work-family balance: "4 kids home schooling. . .with huge learning differences. . . chaotic house. . .. Both of us essential workers (also picking up additional hours to support the COVID demands) but some fear regarding keeping kids at school when advice is to keep home if you can . . .elderly neighbours in quarantine from cruise ship so shopping for them as well. Lot of products unavailable and quantity restrictions in supermarkets is stressful for large family, in addition to queuing. . . ever changing information and advice. Poor internet reliability, not enough technology for all members of family to be schooling on-

line at once, data insufficient for additional use at home. Updating [my]Will. Not enough hours in the day to get everything done resulting in less sleep." (T256)

Nurses and midwives were always cognisant of the potential of viral transmission from the workplace to the home and were implementing new home cleaning routines to try and establish barriers that separated them from their family. They described physical barriers and self-isolating from family to prevent transmission from work to their loved ones.

Nurses and midwives commented at length about the isolation from family locally, interstate and around the world, from saying goodbye to dying relatives and attending funerals. They too were struggling to find ways to support family members with high needs: "I care for my aged mother who lives on her own with dementia. The support groups are no longer able to provide the same amount of care for her which has added more pressure on family members. . ." (T2299).

While the COVID-19 pandemic caused much financial hardship and economic downturn throughout the nation with effects felt worldwide, most study respondents felt relatively unaffected as essential workers, grateful to have employment. Many indicated they saved money due to free childcare entitlements, less discretionary spending, and a halt in face-to-face allied healthcare spending. Still, nurses and midwives endured the economic effects of widespread unemployment of family members or family business and superannuation investment losses. Loss of the second incomes resulted in mortgages put on hold. Many described feeling fortunate they could financially support their family. Those facing immediate retirement indicated the loss associated with superannuation investments would impact their retirement plans, having to extend their working life.

Changes to pandemic related leave entitlements occurred initially causing confusion with the lack of clarity around leave while waiting for test results. Annual leave and long service leave entitlements were used to supplement sick leave, with nurses and midwives describing a system of leave entitlements which was ill-fitting to the needs of the COVID-19 pandemic.

**Theme 4: Workforce dissolution–Subtheme: Disengaged.**   The chronic shortage of nurses and midwives was compounded as workforce numbers dropped further with health services immediately changing their model to re-direct resources to the emergency pandemic cause. However, absenteeism of health care staff placed additional demands on the remaining workers, leading to increased workload, unsafe staffing levels and in turn increased sick leave. Nurses and midwives experienced rapid dissolution of their workforce: "the thing that has impacted me most at work is that others are not coping—huge impact on absenteeism across the workforce" (T3289). All planned staff leave was cancelled. Nurses and midwives indicated the work stress was causing sleep issues, with many concerned about the plight of colleagues intrastate and interstate. Nurses and midwives explained their frustration and dismay not meeting patient and family expectations and failing to provide quality patient care: "It was really hard as some nurses were coming to work but lots weren't. This meant we worked down. Yesterday on the late shift, we normally run with 11 nurses, we had only 5 and 3 of them were doing doubles. Families get angry with us for not giving their loved one enough care. This happens day after day and we're getting so tired" (T1451).

Increasing capacity meant changes to hospital infrastructure seeing the relocation and redesign of acute wards and services. Increasing human resources is always a priority in nursing and April 2020 saw the commencement of nursing graduates to the workforce, adding to the stress of existing staff and making the transition of graduates very challenging. Nurses and midwives described setting their colleagues up to fail in a system that could not provide adequate transition support: "Approx 130 Graduate nurses have started, how are these already stressed young nurses at the start of their career meant to feel supported, when the current staff feel the ongoing stressors and struggle to come to work" (T3561). The pressure for the

Transition nurses was stressful, and intense: "I'm mature aged and a late starter in nursing. One of the most stressful careers I have ever had and not sustainable" (T229). Describing low morale, nurses and midwives became professionally disengaged and "presenteeism".

Many nurses and midwives were forced to change and expand their role and responsibilities. As intensive care departments were put under pressure, staff from other areas were told they may be redeployed to assist in these areas. These 'surge' nurses described training as rushed, stressful and inadequate, seeing staff from acute and subacute areas completing short online education with no associated hands-on clinical training. Those completing the on-line surge training were concerned about their scope of practice and the impact on their professional registration while leaving the experienced intensive care nurses feeling devalued, fearing standards of patient care would not be maintained and raised the issue of building capacity for critical care nurses in the future: "This training devalues all the experienced critical care nurses....I only have 3 years of experience and still feel junior...it takes years and years of experience to work here" (T3455).

Many highlighted problems managing their existing workload, already feeling pushed to the brink of burnout, expressing feelings of failure and doubt about their capability to take on more while others expressed guilt if unable to work. There were grief reactions in response to the chaotic losses taking place. Nurses and midwives worried the current system did not have the surge capability or capacity to successfully manage the COVID-19 pandemic. They describe a failing health system prior to COVID-19: "The current stress of nurses, staff and colleagues within the acute care facilities at this time does not stem from the COVID pandemic directly, but indirectly. The stress and workload demand that stems from the influx of highly acute patients and high level of patient flow and bed block is now daily life or normal..." (T1412).

**Theme 5: Positive life change–Subtheme: Certainty.** Lastly nurses and midwives wrote at length about the positives of the COVID-19 pandemic. They described positive benefits, particularly in home and family relationships, more time to focus on reading and gardening and valuing their job security. Nurses and midwives never lost sight of the difficulties their colleagues were enduring locally and internationally and considered themselves very grateful to have a lifestyle of relative freedom and economic and health security: "I reduced my work hours and undertook value reflection as a result of COVID. My quality of life and feelings of happiness have improved as a result" (T3321). By Survey 3, nurses and midwives were working towards the COVID-19 vaccine roll out which provided a focus of certainty, positivity and end to the pandemic: "We are more financially secure now and more connected as a family than ever. Home-schooling works for us, our mortgage is on hold for 6 months and my partner is currently being supported with a Jobseeker payment. ...now these things are in place we are not stressed." (T3166).

## Discussion

As the largest cohort of health professionals, nurses and midwives around the world were thrust into the public eye as they 'waged the war' against COVID-19.

Overall psychological distress prevalence estimates are higher than those seen in the Australian general population, although considerably lower than those among nurses described in a recent meta-analysis [5]. We showed in Surveys 1, 2 and 3 respectively diagnostic levels of anxiety (GAD-7 ≥10: 23%, 18%, 21%), depression (PHQ-9 ≥10: 26%, 23% and 28%), PTSD (IES-R ≥37: 16%, 12% and 10%) and insomnia (ISI ≥22: 19%, 19% and 21%), whilst 6.7% of nurses and midwives displayed severe levels across all outcomes. Symptoms of severe anxiety

and depression increased over the year, whilst PTSD-symptoms diminished, and insomnia remained steady.

This study was accessible to all public sector nurses and midwives in Tasmania. Forty-six percent of study respondents were aged greater than 51 years, an older sample compared with the average age of nurses in Australia of 44 years [40]. The two most prominent associations with psychological distress were home and family stress, and poor workplace support from colleagues and managers. The former association appears to have decreased over the year of the survey, whilst the latter became the strongest association at the end of the year. Also, increased age is associated with reduced levels of anxiety and depression at all survey periods. Similarly, 52% of the study cohort hold a graduate certificate or higher which is also associated with a reduction in levels of insomnia and general stress. Finally, 12% of the study sample identify as male, which was also found to be associated with lower levels of anxiety, depression and general stress. While it is recognised that men are far less likely to admit to themselves they have a mental health issue or seek help than women [41] underlying symptomology may potentially be underestimated in this subgroup.

The immediate concern for nurses and midwives as the pandemic unfolded was disease transmission prevention and mitigating spread. The focus quickly became personal protection; the correct, efficient, and effective use of PPE. Nurses and midwives articulated PPE as a source of stress: access to PPE and the correct use of PPE, which is consistent across the literature [20]. Nurses and midwives concern about adequate access to PPE was reflected within general stress and insomnia domains. Free-text comments revealed the organisation's strategy to provide a PPE training video for consistent education and training became both a stressor and support. Working in wards, corridors or medication/treatment rooms and sharing staff tea rooms and communal areas where reducing staff density was not feasible, represented double standards for nurses and heightened their risk of exposure and transmission. The use of PPE for other healthcare associated infections were reduced so PPE could be conserved for COVID-19 patients.

While actual numbers of COVID-19 cases were low in Tasmania, there remained a constant stream of suspected COVID-19 admissions throughout Tasmanian hospitals as the state-supported repatriation of international and interstate residents, essential workers including seasonal workers and border re-openings. Nurses and midwives working in COVID-19 designated areas were affected by the constancy of the increased workload, and constant threat of COVID-19.

Levels of stress, anxiety, depression, and insomnia relating to the current family and home situation were high with levels continuing to climb throughout all surveys. Respondents free text spoke about the impact of the pandemic on their social and family lives. This study shows nurses and midwives shared similar fears to the general population. They feared the unknown, feared for their family and significant others which became heightened with border closures, preventing many accesses to their friends and family's interstate and internationally for long periods. Many were impacted by family bereavement, unable to attend funerals or celebrate family events. This effect was evident at all time points as the strain of lack of contact continued to be felt, triggering a process of grieving and a sense of loss of missed time and opportunities. Many nurses felt at risk, stating 'they did not sign up for this' describing unacceptable levels of risk for healthcare professionals during the pandemic. However, overall anxiety and PTSD became less associated with current and future family stress during the year.

The effect of COVID-19 caused sudden financial and economic instability and was a time of great stress [42]. Current and future financial stress was evident among nurses and midwives with associated high levels of insomnia and general stress. Their comments showed that many nurses and midwives and their families suffered financial hardship, and were very

grateful to have assured employment, while others felt financially stronger though economically vulnerable.

Lack of clear communications and advice greatly contributed to the stress felt by responders as they attempted to make sense of their rapidly changing world. 'Fear spreads even more quickly than infections,' [43] particularly true in Surveys 1 and 2. Nurses and midwives indicated significant levels of stress, anxiety and depression related to inadequacy of information and the misinformation generated from both the general and social media just as the pandemic was beginning to bite. During this time, and equipped with very little pandemic preparedness, there was considerable role change which included the formation of the COVID-19 emergency-response management structure. Initially, information flow seemed blocked to staff on the frontline, with many nurses and midwives not aware of changing directions and planning. Nurses and midwives were overloaded with information from many sources causing information fatigue. Professional employment issues, such as pandemic leave and associated changes to leave entitlements were also rapidly changing, leaving nurses and midwives without clear scope of employment boundaries.

Past studies looking at nursing culture [44] show nurses and midwives place great value in workplace team support and the pandemic has shown two extremes; both nurses and midwives working together and the other extreme of feeling alone, isolated, and unsupported. Losing workplace team support was of great concern for nurses and midwives, with results showing high levels of anxiety, depression, insomnia and general stress relating to concern for adequate team support increasing across the study period. Workplace team support has been shown to have direct effect on patient safety and missed care [45] and burnout [46]. Given the downturn of the perceived threat of COVID-19 at the Survey 3 with high levels of distress with an increasing trajectory across all domains, highlights the need for further research to understand the ongoing causes of stress and mental exhaustion in the organisational domain. Australian Institute of Health and Welfare acknowledge burnout as part of the 11th Revision of the International Classification of Diseases (ICD-11-AM) [47]as an occupational phenomenon resulting from chronic workplace stress, not successfully managed. Study data uncover outcomes which are considered risk factors associated with a pre-disposition to burnout.

Respondents' comments indicate exhaustion stemming from working long hours with unrelenting pressure which has seen to precipitate both absenteeism [48] and presenteeism [49] due to high mental distress rather than physical unwellness. Nurses and midwives often put the needs of the team before their own and hold a cultural ideal of team loyalty [50]. Nurses and midwives will come to work when they are sick to not let the team down. While COVID-19 screening for respiratory illness may have reduced the rate of presenteeism from physical unwellness [51], results show high levels of presenteeism among Tasmanian nurses and midwives suffering mental distress.

As a profession, nurses and midwives are altruistic [52] having a strong sense of doing the best they can with limited resources available [53]. We saw nurses and midwives still going to work, under greater pressure with the desire to do 'what was right' for the patients; struggling to meet their own high expectations of quality patient care yet unwittingly subjecting the patients to high risk of adverse patient safety events [54]. At times the sense of not being able to do enough was heightened when nurses felt alone and at times in a professional void without an end in sight. Free text comments show the words 'burnt out', 'disengaged' with nurses questioning their loyalty for the profession into the future. Nevertheless, the additional demands placed on nurses and midwives during the pandemic were not abstract, they were very real. Nurses described ethical stress from practice change; forced to adopt practices which did not adhere to best practice due to social distancing (such as changes to clinical handover, ward rounds and team meetings, meal breaks, management, and committee meetings). For

nurses and midwives, the imperative of providing high-quality patient care is very highly valued, integral to their professionalism. The responsibility for balancing quality of care with availability of resources rests firmly with senior political and professional managers of the health service. In practical terms, for nurses and midwives, there was little recognition of that responsibility, and no explanation of how they might be going about managing this balance. As a result, the pressure-points were transferred to the frontline workers, who were unable to handle the balancing of the strategic planning processes.

Health services normally operate in a climate where there is always more demand than can be satisfied. The COVID-19 pandemic placed massive additional demand on nurses and midwives particularly in public hospitals, taking their pre-existing high stress to extreme levels or overload. Health service management responded by cutting-back the provision of other services, however this resulted in the move from familiar practice to unfamiliar practice. To a variable extent, management was aware of this and has responded with varying degrees of skill and comprehension.

By 12-months, surge plans and capacity preparedness became part of the norm. The World Health Organisation provided guidelines for organisations [55], reinforcing that as the pandemic progressed, more information and scientific knowledge will be gained, and the response enacted accordingly. Among emergency service operations and planning, the question arises as to the adequacy of plans with respect to dealing with the psychological effect of demand changes, and whether the line managers were adequately trained and appropriately selected to implement the plans that were devised. Even if the emergency plans were appropriately designed, the question remained around what to do about the care burden of postponed services, and the nagging awareness of that deficit in service among the health professionals who were required to change priorities.

High personal expectations enhanced by a sense of duty, coupled with professional and personal demands placed on nurses by the pandemic created personal discord. It was evident by Survey 2 that nurses and midwives were being confronted with anti-social public behaviour and recommended to wear plain clothes when travelling to and from work to minimise the risk of abuse. Such behaviour placed even more stress on nurses and midwives, the profession who are protectors, now requiring protecting and measures to look after themselves in order to care for the patients. Nursing and midwifery are stressful professions [56], and stress may originate from the working environment, individual, or organisational factors, or a combination of these.

This study was very sensitive to the possibility questions may precipitate existing stress among respondents. Levels of extreme stress with thoughts of self-harm and feeling as though there was no end, and no way out were uncovered in study responses. Extreme levels of stress have the potential to manifest as PTSD which need to be addressed by organisations. The study survey provided links to support services for respondents.

## Study limitations

An anonymous online survey is not capable of producing an unbiased estimate of the prevalence of health conditions, since it is unknown whether psychological conditions may increase or reduce the likelihood of a person responding to the survey. It is also unknown how many people will have received the survey request as the email list may not be up to date resulting in emails not reaching participants.

This study did not measure physical health and symptoms of nurses and midwives. There is a complex relationship between psychological distress and physical symptoms where psychological distress exacerbates physical symptoms and vice versa and the study cannot differentiate physical distress as a predictor.

It was not possible to perform reliable repeated-measures analyses, as identification of individuals did not appear to be accurate, thus increasing the 95% confidence intervals of outcome estimates. Commercial online survey tools are not able to provide reliable linkage of individual's responses whilst maintaining anonymity. Thus the design became a hybrid longitudinal study. Nevertheless, this does not invalidate the analysis performed and conclusions drawn in our opinion.

The longevity of the pandemic was underestimated. Indefinite extension of the study would provide a better description of the long-term psychological health of nurses, and how the pandemic interacted with pre-existing organisational factors.

## Conclusion

Entering a pandemic unprepared proved to be a test for the effectiveness and morale of the community and health system. Primary findings include:

Workplace team support directly affected stress and mental exhaustion among nurses and midwives and a strategic priority should include building and strengthening the workplace team, supports and culture into the future.

The impact of COVID-19 on current and future nurses' and midwives' home and social lives caused high levels of stress and mental exhaustion. While the provision of stable and functional relationships at home cannot be controlled by the health service, there may be opportunities to provide targeted support (e.g. in domestic violence situations). Understanding strategies to provide support at home for nurses and midwives may provide protection, social care, improve wellbeing and strengthen the culture of valuing nurses and midwives.

Lack of organised and responsive communication processes impacted nurses and midwives during the COVID-19 pandemic. The initial three months of COVID-19 was particularly difficult for nurses and midwives, dealing with the rapid changes with little direction. Processes for clear communication would help to mitigate confusion and distress in the future.

PPE availability issues caused much stress and mental exhaustion. Nurses and midwives regard being protected at work as a priority for their and their family's safety, reflecting the level of respect and regard (or lack thereof) the organisation had for them as professionals. Managing organisational risk and safety of all healthcare workers is a legal obligation for the organisation, ensuring readily available and accessible PPE stock, and adequate staff training.

Younger nurses and midwives suffered higher levels of stress and mental exhaustion. Fostering resilience may help support the future nursing and midwifery workforce rather than fuelling attrition.

## Recommendations for further research

Embedding future research to examine levels of burnout among nurses and midwives will aid workforce sustainability.

## Supporting information

**S1 File.**
(PDF)

**S2 File.**
(ZIP)

**S3 File.**
(DOCX)

## Author Contributions

**Conceptualization:** Kathryn M. Marsden, J. Porter.

**Data curation:** Kathryn M. Marsden, I. K. Robertson.

**Formal analysis:** Kathryn M. Marsden, I. K. Robertson.

**Funding acquisition:** Kathryn M. Marsden.

**Investigation:** Kathryn M. Marsden, I. K. Robertson.

**Methodology:** Kathryn M. Marsden, J. Porter.

**Project administration:** Kathryn M. Marsden.

**Resources:** Kathryn M. Marsden.

**Validation:** Kathryn M. Marsden, I. K. Robertson.

**Writing – original draft:** Kathryn M. Marsden.

**Writing – review & editing:** Kathryn M. Marsden, I. K. Robertson, J. Porter.

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
