## [Decision Letter · Decision Letter 0]

11 Feb 2022

PONE-D-21-36897Stressors, Manifestations and Course of COVID-19 Related Distress Among Nurses and Midwives in TasmaniaPLOS ONE

Dear,

Thank you for submitting your manuscript to PLOS ONE. After careful consideration, we feel that it has merit but does not fully meet PLOS ONE’s publication criteria as it currently stands. Therefore, we invite you to submit a revised version of the manuscript that addresses the points raised during the review process. Please submit your revised manuscript by 10th March 2022. If you will need more time than this to complete your revisions, please reply to this message or contact the journal office at plosone@plos.org. Please include the following items when submitting your revised manuscript:A rebuttal letter that responds to each point raised by the academic editor and reviewer(s). You should upload this letter as a separate file labeled 'Response to Reviewers'.A marked-up copy of your manuscript that highlights changes made to the original version. You should upload this as a separate file labeled 'Revised Manuscript with Track Changes'.An unmarked version of your revised paper without tracked changes. You should upload this as a separate file labeled 'Manuscript'.

We look forward to receiving your revised manuscript.

Kind regards,

Muhammad Shahzad Aslam, Ph.D.,M.Phil., Pharm-D

Academic Editor

PLOS ONE

Journal Requirements:

"No"

Reviewers' comments:

Reviewer's Responses to Questions

**Comments to the Author**

1. Is the manuscript technically sound, and do the data support the conclusions?

Reviewer #1: Yes

Reviewer #2: Partly

2. Has the statistical analysis been performed appropriately and rigorously? 

Reviewer #1: Yes

Reviewer #2: No

3. Have the authors made all data underlying the findings in their manuscript fully available?

Reviewer #1: Yes

Reviewer #2: No

4. Is the manuscript presented in an intelligible fashion and written in standard English?

Reviewer #1: Yes

Reviewer #2: Yes

5. Review Comments to the Author

Reviewer #1: PONE-D-21-36897

Thank you for your contribution to the PLOS ONE.

This study describes the critical stressors and how they manifest within both the work and larger social environment for nurses and midwives in Australia.

I have following comments to the authors

Abstract

There seems to be a lack of definition in the text of organizational burnout. The Manuscript contains few statements on nurses’ burnout. As burnout was not directly measured in this study, it is necessary to provide an additional explanation about the relationship between burnout and psychological distress investigated in this study

Page 3, the full terminology of PTSD is required.

Method

Page 6, participants – It needs additional information on how many people were surveyed and what the response rate was.

Page 8, line 196-7 & page 9, line 205-207 – They seem unnecessary as measurement descriptions

The description of qualitative data collection is somewhat lacking, i.e. what questions were used, participants, setting, etc.

Page 10, Qualitative analysis – Please add a reference for the thematic analysis method that you used

Results

Overall, a description of the results based on the results presented in the table is required.

Table 1, survey 2 – There is an error on the percentage of group 51+.

Table 2, the classification ranges of ISI and IES-R are different from the description on page 8.

Please cite figures as “Fig 1”, “Fig 2” at the end of explanation, etc. as per PLOS ONE manuscript guideline.

Page 17, line 380-381 box plots 5A…. => 4A….

Figure 5 – Please consider elaborating or deleting the current (and future) financial stress as it seems to lack justification to view them as COVID-19 related psychological distress.

Page 19 – line 402-403 box plots 6A…. => 5A….

Since it seems unnecessary to duplicate tables and figures on the same subject, it is recommended to present only one.

Page 21 – In Theme 1, 2, and 5, it is difficult to guess who the subjects of the content are and whether the content is positive or negative, so it is necessary to modify the theme names so that the direction is presented. In addition, please consider to provide sub-categories therefore it is possible to understand how these themes were derived from the qualitative data.

On page 21, theme #6 (theme #4 on page 27) is missing, and it does not match the theme presented on page 27-29.

Theme 1 – It is little difficult to understand due to lack of explanation in the text on how communication and planning are grouped into one topic.

Page 22 Line 460-462 - It is difficult to see this as a problem due to the lack of advice related to COVID-19.

Page25 line 529- It seems inappropriate to mention the patient's vulnerability and the nurse's vulnerability within the same theme,

Line 533-537, this statement does not appear to be a difficulty with the participant's (nurse) work or COVID-19 related Psychological Distress.

Research questions about qualitative analysis are not described in the research purpose and methods. In the 5-6 derived themes, in particular, planning and accessibility of care do not seem to be the best analysis results as COVID-19 related psychological distress.

Discussion

Discussions should be based on the main findings of the study. For example, line 685- on page 31 is not seen as a discussion based on the study results. Line 706-, the information related to mandatory vaccination was not presented in the result section.

Line748- It is not clear which theme is related to burnout in the qualitative analysis results.

Conclusion needs be described more compactly..

Line 847- Recommendations for further research – Please describe your specific research proposal in sentence format.

Since the manuscript is generally too long, it needs to be edited, and it seems appropriate to move some of the content to the discussion or delete (eg, line 808-).

It is a meaningful study as it saw the longitudinal effect of COVID-19, but it seems that the advantages of longitudinal data are not sufficiently reflected and sufficiently discussed.

Reference #10, 27, 55 - Organisation Wh => World Health Organisation

Reviewer #2: 1. The method used to treat missing data in this study is mean substitution, but it tends to reduce variability, increase R^2, and decrease SE by design. According to Dr. Paul Allison at the University of Michigan, “All of these [single] imputation methods suffer from a fundamental problem: Analyzing imputed data as though it were complete data produces standard errors that are underestimated and test statistics that are overestimated. Conventional analytic techniques simply do not adjust for the fact that the imputation process involves uncertainty about the missing values.” It is strongly recommended to use other advanced methods, such as multiple imputation (MI) or full information maximum likelihood (FIML), to handle missing data. FYI, in many situations, these methods can also handle missing data in outcomes as well. Or even listwise deletion might be a better option than mean substitution.

2. Please provide the missing data information (N valid and invalid for each variable) for all variables used in this study.

3. How the outcomes (PHQ-9, GAD-7, ISI, IESR-22) were treated for analysis is unclear. The authors stated that “The four primary outcome scales derived from the battery of psychological tests (PHQ9, GAD7, ISI, IESR 22) are rank ordered with relatively broad ranges but are USUALLY TREATED as continuous interval variables for purposes of statistical analysis.” It seems like they were treated as continuous (Table 3) AND ordinal (Tables 4 and 5), but it is unclear what it means by “usually treated as continuous interval variables”.

4. How the longitudinal data was analyzed is confusing and unclear. There are three-time points, and the outcomes can be treated as continuous or ordinal. There are four outcomes, hence there should be 8 models (=4 outcomes continuous and ordinal) at least. Then a linear mixed model can be used with a continuous/ordinal outcome and time as a fixed effect controlling for the covariates. In each model, the “trend” can be shown by testing the interaction between time and a predictor. Then marginal predictions can be calculated by using the program based on the model estimates along with estimate/SE/95% CI, not observed values (mean/SD). Figures need to be based on these values. Also, it seems that the data from three times were merged (Tables 4 and 5) for some reason then analyzed, while they could have been handled within a linear mixed model regardless of the nature of the outcome (continuous or ordinal).

5. Please consider treating the outcomes as either continuous or ordinal, not both. It is a matter of how each outcome needs to be handled to better answer the research questions of the study. It is a technical as well as theoretical question that need to be answered before running any statistical model.

6. Please provide more details about each linear mixed model, such as model fit (e.g., AIC/BIC), sample size, the statistical method for fitting the model (e.g., ML or REML), the structure of the covariance matrix for the random effects, and the structure of the residual errors within the lowest-level groups. None of these are reported.

7. Study hypotheses are implied but not clearly stated at all in the introduction.

8. On line 21, please provide the details about “3 timepoints over 12-month”: baseline / 3-month / 12-month.

9. The terms used to describe the statistical model on line 26-27 and 215-216, “mixed effects linear regression and linear mixed model analyses” and “mixed effects linear regression and mixed effects ordered logistic regression”, are very confusing and somewhat incorrect. The statistical method used for analysis is a linear mixed model (aka multilevel model) with continuous and nominal outcomes. Or it can be specified as “multivariate mixed effects ordered logistic regression” as the authors specified on line 223-224.

10. On line 144-146, please discuss how the exclusion of the North West region hit hard by the pandemic would have affected the data and the results. They seem to be the one that might have been most impacted by the pandemic but missing from the data.

11. Please add more details about how demographic and “other relevant” (please specify) predictor variables were measured (line 149-157). For example, it is unclear how “education level” or “community exposure” was measured by the survey. They are somewhat described in tables, but the authors should provide much more details about the variables used for analysis with text.

12. Please specify what measure (e.g., Cronbach’s alpha) was used for “internal consistency”.

13. Line 210-211 is not necessary. Please consider removing the sentence.

14. Please add the total N for each variable in Table 1.

15. In Tables 4 and 5, the outcome variables are listed as predictors. Please reorganize the tables.

6. PLOS authors have the option to publish the peer review history of their article (what does this mean?). If published, this will include your full peer review and any attached files.

Reviewer #1: No

Reviewer #2: No

---

## [Author Response · Author response to Decision Letter 0]

28 Mar 2022

Please refer to the rebuttal letter for authors comments. There are too many to list here as it was a major revision. In short, there was a complete statistical reanalysis and formation of new accompanying tables and figures.

I have attempted to update the 'Competing Interests' section more comprehensively as required, however the section is not available for completion at time of submission.

However, "the authors have declared that no competing interests exist".

---

## [Decision Letter · Decision Letter 1]

12 Apr 2022

PONE-D-21-36897R1Stressors, Manifestations and Course of COVID-19 Related Distress Among Public Sector Nurses and Midwives during the COVID-19 Pandemic First Year in Tasmania, AustraliaPLOS ONE

Dear Dr. Marsden,

Thank you for submitting your manuscript to PLOS ONE. After careful consideration, we feel that it has merit but does not fully meet PLOS ONE’s publication criteria as it currently stands. Therefore, we invite you to submit a revised version of the manuscript that addresses the points raised during the review process.

We look forward to receiving your revised manuscript.

Kind regards,

Muhammad Shahzad Aslam, Ph.D.,M.Phil., Pharm-D

Academic Editor

PLOS ONE

Additional Editor Comments:

Please correct the mistakes given by reviewer and resubmit for further review

Reviewers' comments:

Reviewer's Responses to Questions

**Comments to the Author**

1. If the authors have adequately addressed your comments raised in a previous round of review and you feel that this manuscript is now acceptable for publication, you may indicate that here to bypass the “Comments to the Author” section, enter your conflict of interest statement in the “Confidential to Editor” section, and submit your "Accept" recommendation.

Reviewer #1: All comments have been addressed

Reviewer #2: (No Response)

2. Is the manuscript technically sound, and do the data support the conclusions?

Reviewer #1: Yes

Reviewer #2: Yes

3. Has the statistical analysis been performed appropriately and rigorously? 

Reviewer #1: Yes

Reviewer #2: Yes

4. Have the authors made all data underlying the findings in their manuscript fully available?

Reviewer #1: Yes

Reviewer #2: Yes

5. Is the manuscript presented in an intelligible fashion and written in standard English?

Reviewer #1: Yes

Reviewer #2: (No Response)

6. Review Comments to the Author

Reviewer #1: Since this manuscript is rather lengthy for the reader to read, it is recommended to make the manuscript more compactly.

Reviewer #2: I really appreciate time and effort to address the comments and suggestions with details. The manuscript has a lot of valuable and precious information for a variety of readers in the context of pandemic. A few additional comments and suggestions.

I agree that not many will be interested or have understanding in the technicalities of the statistical analysis used for this manuscript. However, I believe it is important to report technical details as much as possible since it is a peer-reviewed academic manuscript. And the manuscript is already very technical with many numbers to interpret. I see that a lot of information is provided as supplement, but it would be very informative if some key information at least can be added in the main text. For example, key details about multiple imputations, such as number of imputations (and justification) and auxiliary variables used if used any. How some assumptions were violated, what command was used in Stata, and/or what was not available in the Stata version you used, are all important but not essential, hence they can be placed in the supplement or briefly mentioned in text in my opinion.

Removing or including variables in the model is more of a matter of theoretical considerations, not significance, in my opinion. The author(s) reasonably explained the decision about variable choices. It is impossible for observational data to measure everything for sure, and the author(s) appropriately went through various technical and theoretical considerations in depth.

In general, -2LL, AIC, and BIC are the trios of model fit when it comes to longitudinal data analysis, and they do not always agree. FYI, AIC weight number of parameters more compared to BIC while BIC takes N into the calculation as well as number of parameters. My suggestion is reporting the range of model fits based on the analysis results with the multiple imputation datasets. Perhaps, the model fits without using MI might be preferred to range. Yes, no command is available yet when multiple imputation command (mi) was used in Stata, but it does not mean that no model fit needs to be reported.

Please explain why time was treated as numeric (.1, .29, and 1.1 years) instead of categorical. There are only three time points. How time was treated in the current models assume a linear trend over time.

Please explain how the online data collection was done with details. It is unclear if the study used a online data collection platform, such as REDCap, Qualtrics, QuestionPro, or else, or a document was attached to the email sent and participants sent the completed surveys back to the researchers.

Please consider merging “Survey timepoints” with “Study design”.

Please be consistent with the terms for the surveys. It is unclear if “survey one or survey 1”, “survey two or survey 2”, and “survey three or survey 3” mean “survey” at baseline, 3-month, and 12-month, respectively.

The author(s) explained that Survey one was only distributed to Southern Tasmanian region, and Survey two was distributed only to the Southern and Northern regions, while Survey three was distributed to Southern, Northern, and Northwestern regions. It seems like only those who were in the Southern region could complete the surveys at all three time points, which means that 1) the participants from the other two regions have missing data for the survey at baseline (Survey one), and that 2) the participants from the Northwestern region have missing data for the surveys at baseline and 3-month. It would be very information if the author(s) can share some insight about how this might have impacted the representativeness of the data compared to the general population of nurses and midwives in Australia.

In Figure 1, please add the month of the first survey (April).

7. PLOS authors have the option to publish the peer review history of their article (what does this mean?). If published, this will include your full peer review and any attached files.

Reviewer #1: No

Reviewer #2: No

---

## [Author Response · Author response to Decision Letter 1]

20 May 2022

I have addressed all reviewer's comments in the rebuttal letter and provided all additional analysis and supporting documents. I trust this will now be satisfactory. Thank you.

---

## [Decision Letter · Decision Letter 2]

20 Jun 2022

PONE-D-21-36897R2Stressors, Manifestations and Course of COVID-19 Related Distress Among Public Sector Nurses and Midwives during the COVID-19 Pandemic First Year in Tasmania, AustraliaPLOS ONE

Dear Dr. Marsden,

Thank you for submitting your manuscript to PLOS ONE. After careful consideration, we feel that it has merit but does not fully meet PLOS ONE’s publication criteria as it currently stands. Therefore, we invite you to submit a revised version of the manuscript that addresses the points raised during the review process. Please submit your revised manuscript by 20th July 2022. If you will need more time than this to complete your revisions, please reply to this message or contact the journal office at plosone@plos.org. Please include the following items when submitting your revised manuscript:A rebuttal letter that responds to each point raised by the academic editor and reviewer(s). You should upload this letter as a separate file labeled 'Response to Reviewers'.A marked-up copy of your manuscript that highlights changes made to the original version. You should upload this as a separate file labeled 'Revised Manuscript with Track Changes'.An unmarked version of your revised paper without tracked changes. You should upload this as a separate file labeled 'Manuscript'.If applicable, we recommend that you deposit your laboratory protocols in protocols.io to enhance the reproducibility of your results. Protocols.io assigns your protocol its own identifier (DOI) so that it can be cited independently in the future. For instructions see: https://journals.plos.org/plosone/s/submission-guidelines#loc-laboratory-protocols. Additionally, PLOS ONE offers an option for publishing peer-reviewed Lab Protocol articles, which describe protocols hosted on protocols.io. Read more information on sharing protocols at https://plos.org/protocols?utm_medium=editorial-email&utm_source=authorletters&utm_campaign=protocols.

We look forward to receiving your revised manuscript.

Kind regards,

Muhammad Shahzad Aslam, Ph.D.,M.Phil., Pharm-D

Academic Editor

PLOS ONE

Journal Requirements:

Reviewers' comments:

Reviewer's Responses to Questions

**Comments to the Author**

1. If the authors have adequately addressed your comments raised in a previous round of review and you feel that this manuscript is now acceptable for publication, you may indicate that here to bypass the “Comments to the Author” section, enter your conflict of interest statement in the “Confidential to Editor” section, and submit your "Accept" recommendation.

Reviewer #2: All comments have been addressed

Reviewer #3: (No Response)

2. Is the manuscript technically sound, and do the data support the conclusions?

Reviewer #2: (No Response)

Reviewer #3: Yes

3. Has the statistical analysis been performed appropriately and rigorously? 

Reviewer #2: (No Response)

Reviewer #3: Yes

4. Have the authors made all data underlying the findings in their manuscript fully available?

Reviewer #2: (No Response)

Reviewer #3: Yes

5. Is the manuscript presented in an intelligible fashion and written in standard English?

Reviewer #2: (No Response)

Reviewer #3: Yes

6. Review Comments to the Author

Reviewer #2: I appreicate the time and effort the author(s) have spent to address my comments. I have no further comemnts.

Reviewer #3: Please provide numbers for participants in each wave in the section on participants. It is not clear in that section if the same people responded to each wave, or new participants were in each wave. It appears to be more of a panel series design rather than a longitudinal design, which follows the same participants across time.

Please state PHQ 9 as PHQ 2 is commonly used in clinical studies. Provide the range of the Likert scales on line 178. In the section on qualitative data analysis, please provide the questions here.

How was grounded theory methodology modified? (line 201). Was a theory created?

Please explain what an enrolled nurse is and the nurse's educational training for this role. Line 259 is missing a noun, most likely 'data' or 'responses'.

Please explain what Hospital certificate and TAFE certificate means and the educational level for them.

Did you observe or measure the levels of pandemic prevalence during the waves, and did they correspond to the stress experienced by the nurses?

7. PLOS authors have the option to publish the peer review history of their article (what does this mean?). If published, this will include your full peer review and any attached files.

Reviewer #2: No

Reviewer #3: No

---

## [Author Response · Author response to Decision Letter 2]

6 Jul 2022

All queries raised have been considered and documented in the rebuttal letter. Manuscript and supporting materials support any changes. Thank you for your support.

---

## [Editor Report · Decision Letter 3]

8 Jul 2022

Stressors, Manifestations and Course of COVID-19 Related Distress Among Public Sector Nurses and Midwives during the COVID-19 Pandemic First Year in Tasmania, Australia

PONE-D-21-36897R3

Dear,

We’re pleased to inform you that your manuscript has been judged scientifically suitable for publication and will be formally accepted for publication once it meets all outstanding technical requirements.

Kind regards,

Muhammad Shahzad Aslam, Ph.D.,M.Phil., Pharm-D

Academic Editor

PLOS ONE
---

## [Editor Report · Acceptance letter]

18 Jul 2022

PONE-D-21-36897R3 

Stressors, Manifestations and Course of COVID-19 Related Distress Among Public Sector Nurses and Midwives during the COVID-19 pandemic first year in Tasmania, Australia 

Dear Dr. Marsden:

I'm pleased to inform you that your manuscript has been deemed suitable for publication in PLOS ONE. Congratulations! Your manuscript is now with our production department. 

Kind regards, 

on behalf of

Dr. Muhammad Shahzad Aslam 

Academic Editor

PLOS ONE